# Constraints on simulated past Arctic amplification and lapse-rate feedback from observations

Olivia Linke[1], Johannes Quaas[1], Finja Baumer[1], Sebastian Becker[1], Jan Chylik[2], Sandro Dahlke[3], André Ehrlich[1], Dörthe Handorf[3], Christoph Jacobi[1], Heike Kalesse-Los[1], Luca Lelli[4,10], Sina Mehrdad[1], Roel A.J. Neggers[2], Johannes Riebold[3], Pablo Saavedra Garfias[1], Niklas Schnierstein[2], Matthew D. Shupe[5,6], Chris Smith[7,8], Gunnar Spreen[4], Baptiste Verneuil[1,9], Kameswara S. Vinjamuri[4], Marco Vountas[4], and Manfred Wendisch[1]

[1]Leipzig Institute for Meteorology, Leipzig University, Leipzig, Germany
[2]Institute of Geophysics and Meteorology, University of Cologne, Cologne, Germany
[3]Alfred Wegener Institute, Helmholtz Centre for Polar and Marine Research, Potsdam, Germany
[4]Institute of Environmental Physics, University of Bremen, Bremen, Germany
[5]Cooperative Institute for Research in Environmental Sciences, University of Colorado Boulder, Boulder, CO, USA
[6]Physical Sciences Laboratory, National Oceanic and Atmospheric Administration, Boulder, CO, USA
[7]University of Leeds, School of Earth and Environment, Leeds, U.K.
[8]International Institute for Applied Systems Analysis, Laxenburg, Austria
[9]École Polytechnique, Palaiseau, France
[10]Remote Sensing Technology Institute, German Aerospace Centre (DLR), Wessling, Germany

**Correspondence:** Johannes Quaas (johannes.quaas@uni-leipzig.de)

**Abstract.** The Arctic has warmed much more than the global mean during past decades. The lapse-rate feedback (LRF) has been identified as large contributor to the Arctic amplification (AA) of climate change. This particular feedback arises from the vertically non-uniform warming of the troposphere, which in the Arctic emerges as strong near-surface, and muted free-tropospheric warming. Stable stratification and meridional energy transport are two characteristic processes that are evoked as causes for this vertical warming structure. Our aim is to constrain these governing processes by making use of detailed observations in combination with the large climate model ensemble of the 6th Coupled Model Intercomparison Project (CMIP6). We build on the result that CMIP6 models show a large spread in Arctic LRF and AA, which are positively correlated for the historical period 1951–2014. Thereby, we present process-oriented constraints by linking characteristics of the current climate to historical climate simulations. In particular, we compare a large consortium of present-day observations to co-located model data from subsets with weak and strong simulated AA and Arctic LRF in the past to provide different perspectives on both AA and the Arctic LRF. Our analyses suggest that the vertical temperature structure of the Arctic boundary layer is more realistically depicted in climate models with weak Arctic LRF and AA (CMIP6/w) in the past. In particular, CMIP6/w models show stronger inversions at the end of the simulation period (2014) for boreal fall and winter and over sea ice, which is more consistent with the observations. These results are based on observations from the year-long MOSAiC expedition in the central Arctic, together with long-term measurements at the Utqiaġvik site in Alaska, USA, and dropsonde temperature profiling from aircraft campaigns in the Fram Strait. In addition, the atmospheric energy transport from lower latitudes that can

further mediate the warming structure in the free Arctic troposphere is more realistically represented by CMIP6/w models. In particular, CMIP6/w models systemically simulate a weaker Arctic atmospheric energy transport convergence in the present climate for boreal fall and winter, which is more consistent with ERA5 reanalyses. We further show a positive relationship between the magnitude of the present-day transport convergence and the strength of past AA. In the perspective of the Arctic LRF, we find links between changes in transport pathways that drive vertical warming structures, and local differences in the LRF. This highlights the mediating influence of advection on the Arctic LRF and motivates deeper studies to explicitly link spatial patterns of Arctic feedbacks and AA to changes in the large-scale circulation.

# 1 Introduction

The Arctic region has warmed more rapidly than the global average during past decades, which is seen in both observations and model simulations (e.g., Serreze and Francis, 2006; Serreze et al., 2009; Screen and Simmonds, 2010; Polyakov et al., 2012; Stroeve et al., 2012; Wang and Overland, 2012; Cohen et al., 2014). The most recent period of this Arctic amplification (AA) of climate change has started from the end of the 20th century and continues into the 21st century Overland et al. (2008); Serreze and Barry (2011); Wendisch et al. (2023). Several intertwined processes and feedback mechanisms give rise to AA, among these the surface albedo and temperature feedbacks (e.g., Pithan and Mauritsen, 2014; Block et al., 2020). Here we focus on the lapse-rate feedback (LRF) which arises from the vertically non-uniform contribution to the total temperature feedback. This particular feedback contributes at a level that is similar to the surface albedo feedback to AA, but its underlying physical mechanisms are less well understood (Feldl et al., 2020; Lauer et al., 2020; Boeke et al., 2021). Results from the recent multi-climate model ensemble within the 6th Coupled Model Intercomparison Project (CMIP6; Eyring et al., 2016) confirm that the LRF has a unique latitudinal dependence: The multi-model average in Fig. 1a shows a negative LRF in the tropics and large parts of the mid-latitudes, and a positive LRF in the polar regions, primarily die Arctic. Most of the negative feedback contribution comes from the tropical regions, where the warming is amplified in higher altitudes. This enhances the atmospheric long-wave cooling ability towards space.

In the Arctic, the widespread surface-based temperature inversion and limited vertical mixing abilities of the atmosphere cause the major part of the warming to remain in the lower troposphere (Manabe and Wetherald, 1975). This bottom-heavy warming (BHW) is a key feature of the overall positive Arctic LRF (ALRF). Given the muted warming in the free troposphere, the ALRF limits the atmospheric cooling ability in the long-wave radiation spectrum, which ultimately aids the Greenhouse effect. The latitudinal control on the sign of the local feedback makes the LRF an important contribution to AA (Pithan and Mauritsen, 2014; Block et al., 2020).

The ALRF experiences a unique seasonal and spatial variability (e.g., Feldl et al., 2020; Boeke et al., 2021). The major part of the overall positive feedback results from the boreal fall and winter period, where the degree of sea ice retreat has a strong control on the local intensity of the LRF. Local changes in the sea ice concentration are of central importance through changes in the surface turbulent heat fluxes. Primarily the regions with strong sea ice reduction experience an increase in the upward

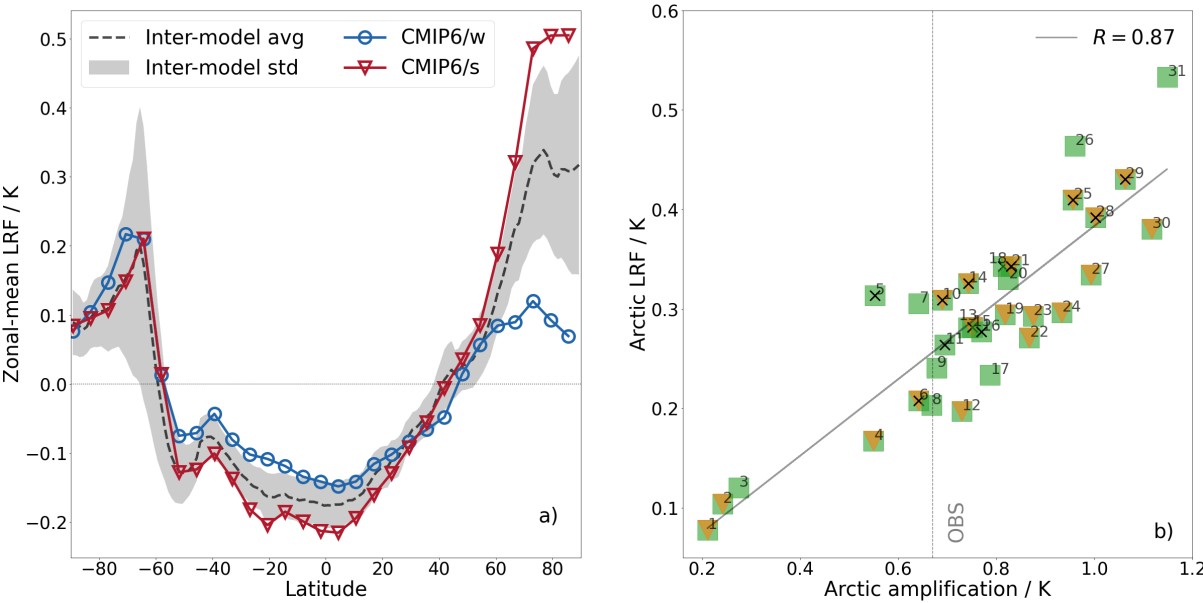

**Figure 1.** a) Zonal and annual-mean LRF (for the period 1985–2014 with respect to 1951–1980) expressed in surface temperature change units, K. The black dashed line indicates the multi-model average (avg) from all 31 CMIP6 models used in this study. The shaded area gives the inter-model standard deviation (std) around the model mean. The blue lines with circles, and red lines with triangles give the average of the 3 models in our collection with lowest, and highest AA (CMIP6/w, and CMIP6/s), i.e. models 1–3, and 29–31 of Tab. 1, respectively. Values for the LRF are derived by calculating the LRF from each radiative kernel (CAM5, GFDL AM2, ERA-Interim, HadGEM3-GA7) and averaging the results. b) Relationship between ALRF and AA in CMIP6 models. As in a), the model-specific temperature change and feedback is derived for the period 1985–2014 with respect to 1951–1980. Green squares represent models for which monthly, orange triangles for which daily, and black crosses for which 6-hourly output data are available from the CMIP6 archive. For the derivation of model-specific AA and LRF, monthly diagnostics of all models (numbering in panel b corresponds to Tab. 1) have been used. Observational estimates (OBS) show the average over AA derived from different observational datasets.

turbulent heating from the ocean, which mediates the local maximum of the winter-time ALRF (Feldl et al., 2020; Linke and Quaas, 2022).

Here, we are interested in the contribution of the LRF to Arctic climate change as observed since 1951. Wendisch et al. (2023) report that in the Arctic (in their study defined as the averaged area north of $60°$ N), the period of 1991–2021 was warmer by $1.33$ K compared to the reference period 1951–1980, which is more than twice the global-mean warming. We

make use of the CMIP6 "historical" simulations with the best estimates of transient climate forcings over the time period of 1850–2014. Here, climate change accounts for the difference between the last 30 years available from the simulations (1985–2014), and an earlier 30 years of the period of interest, 1951–1980. The resulting AA and ALRF are summarised in Fig. 1b, which gives the inter-model spread of AA, which is linearly related to the spread in ALRF (correlation coefficient $r = 0.87$).

We further indicate an observational estimate for AA from several observational data sets. In this study, we define AA as the difference between Arctic (accounting for the area north of $66°$ N) and global-mean warming.

Given the strong seasonal and spatial variability of the ALRF, it is useful to distinguish different seasons as well as different surface-types for a detailed analysis. For the former point, our results are present for different times of the year, depending on the observational constraints. We distinguish boreal spring, summer, fall, and extended winter as April-May-June (AMJ), July-August-September (JAS), October-November (ON), and December-January-February-March (DJFM), respectively. Even though all seasons are considered, we mostly focus on the winter season, where the ALRF is stronger. For the latter point of the surface-control on the ALRF, it is most relevant whether the atmospheric column is over sea ice or open ocean. Since we focus mainly on a model-to-observation comparison over the Arctic ocean, we exclude the influence of snow-covered vs. snow-free land here. It is further relevant for the evolution of the atmospheric temperature profile to distinguish clear and cloudy states of the atmosphere: In the clear state, strong inversions can evolve, and radiative cooling occurs at the surface. With clouds forming, radiative cooling occurs rather in the cloud layer than at the surface, which ultimately weakens the inversion (Pithan et al., 2014).

We firstly motivate the influence of the surface type, and additionally cloudiness on the ALRF during extended winter, purely in CMIP6 data: Figure 2 shows temperature profiles in the lower and middle troposphere, filtered for different conditions. Profiles are categorised into two surface types (sea ice, ocean) and two cloud conditions, based upon a threshold in the total cloud fraction (TCF) within the model grid-cell (TCF $> 99\,\%$, or TCF $\leq 99\,\%$). Therefore, we distinguish four different cases: sea ice / TCF $> 99\,\%$, sea ice / TCF $\leq 99\,\%$, ocean / TCF $> 99\,\%$, and ocean / TCF $\leq 99\,\%$. The sea ice concentration threshold of $15\,\%$ is used to distinguish the sea ice from the open ocean states. The categorisation by cloudiness has the aim to separate the particularly cloudy (overcast) conditions from the rest. We discuss the choice of the TCF threshold later on in the text.

By comparing two different states of cloudiness while considering the same surface type (sea ice or ocean), we at least partly isolate the effect of cloudiness on the temperature profile and its changes. On the other hand, by comparing two different surface types while considering the same state of cloudiness (overcast or non-overcast), we at least partly isolate the effect of different surface types on the temperature profile and its changes.

From distinguishing by surface-type and cloudiness, we motivate observational constraints with the following conclusions from purely model-based outputs:

- Reference and present-day periods: For non-overcast cases (TCF $\leq 99\,\%$), the contrast in surface temperature over sea ice and open ocean dominates the temperature profiles. Over sea ice, strong surface inversions exist while over the relatively warm ocean, the atmospheric boundary layer is well mixed. For overcast cases (TCF $> 99\,\%$), the strong cloud cover reduces the surface temperature contrast between sea ice and open ocean. Over sea ice, cloud top cooling leads to a top-down mixing of the atmospheric boundary layer, which weakens the surface inversion. Some models show a lifted inversion (e.g., CESM2; not shown). Over open ocean, both cloud conditions show a similar stability, but the highly-clouded profile is colder throughout the lower troposphere. This is due to the fact that these cases have their peak in relative occurrence more concentrated towards the sea ice edge (not shown here) compared to the less clouded profiles over ocean.

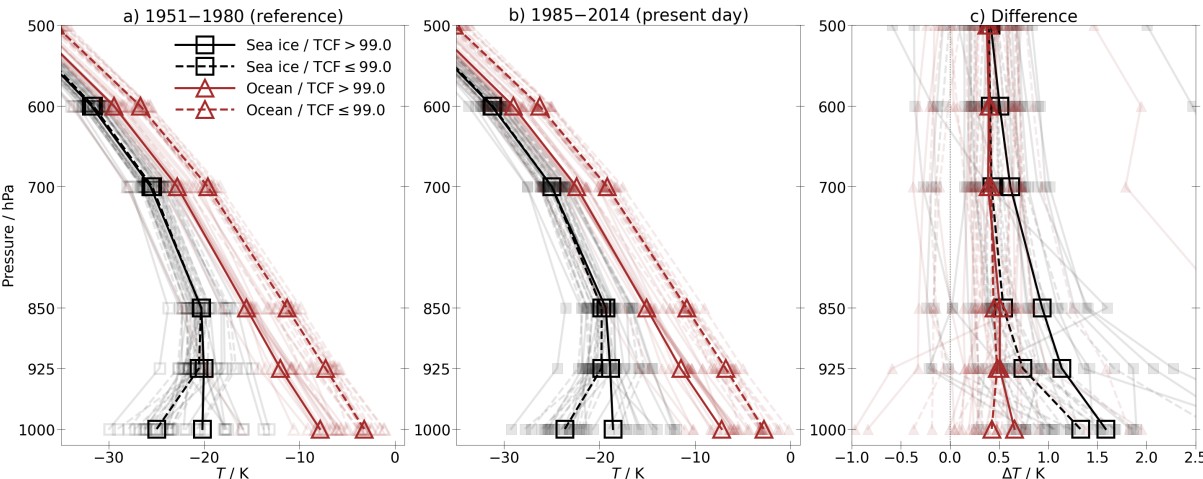

**Figure 2.** Temperature profiles derived from monthly-mean CMIP6 data a) during the reference period 1951–1980, and b) present-day period 1985–2014, as well as c) the difference between the later and earlier period. The season is DJFM. Temperature profiles are derived over sea ice and ocean surfaces for highly cloudy (overcast) conditions with total cloud fractions (TCFs) of TCF > 99 %, and non-overcast conditions, TCF ≤ 99 %, in the columnar monthly-mean files, respectively. The sea-ice concentration threshold is 15 %, above which we define the ocean surface as sea-ice covered. Model grid points are selected as either sea ice or open ocean where these conditions are fulfilled for both reference and present-day period, respectively. The difference profiles in c) are derived from grid points where these conditions are true for both reference and present-day period, respectively. The curves show the multi-model average (thicker curves), and individual models (thin, shaded curves). Note that not all 31 models are included here, as not every model gave an output profile for each of the four classifications by surface-type and cloudiness.

   – Present-day minus reference period: The open-ocean areas show no substantial change in lapse-rate, i.e. no strong LRF results from both cloud conditions over open ocean. However, there is a strong warming near the surface over sea ice for both highly- and less-clouded conditions as compared to over open ocean. The overall warming in the overcast cases is more pronounced than for other conditions, likely due to the fact that these cases appear mostly over the strong-ice melt areas of the Barents-Kara sea (not shown here), which have a notoriously strong warming. However, it is only under overcast conditions that this enhanced warming signal extends up into the mid-troposphere. The gradient of the temperature change from the surface to 850 hPa over sea ice is larger under less-clouded conditions relative to overcast conditions. Thereby, more clouds reduce bottom-heavy warming with respect to the lower troposphere up to 850 hPa. However, considering the entire troposphere (extending from the surface to 300 hPa in our approach), the overall columnar LRF accounting for the lapse-rate change in each layer, is stronger for overcast profiles.

Summarising this introduction to the state of the art, climate models imply a large role of inversion, surface type and clouds for the evolution of the Arctic temperature profile with warming. In addition to that, the thermal structure of the atmosphere can be impacted by remote processes like poleward energy transport. Those controls motivate to investigate if detailed observations or reanalyses can be used as constraints, based upon the CMIP6 inter-model spread in AA and ALRF (Fig. 1b).

The key ideas are:

1. The Arctic LRF is largely controlled by local influences on the surface-near thermodynamic structure: The lack of vertical mixing in the Arctic boundary layer is a key to understand and adequately model the ALRF. As a result, one focus will be on the evaluation of simulated inversion strengths by various means. Additionally, the ALRF is largely depending on the underlying surface type. Most importantly, the strong contrast in LRF and local warming over sea-ice and open-ocean surfaces motivates an evaluation of the simulated warming that is expected through sea ice retreat.

2. The meridional transport of energy in the Arctic free troposphere undergoes a change due to Arctic warming and may amplify or dampen the ALRF by energy advection at different altitudes.

3. The lapse-rate change is linked to cloudiness and vertical mixing strength in the atmospheric column. A further aim is to motivate an assessment of how clouds and boundary-layer dynamics shape changes in the lapse rate by a vertical re-distribution of the warming.

We address point 1 and 2 by comparing present-day (or historical changes in) observations or reanalyses with co-located model data. The constraint is based on the separation of the co-located model data into a subset of models with either weak or strong simulated past AA (and ALRF given their high inter-model correlation; Fig. 1b). By identifying differences between both model subsets, and falsifying one or the other based upon observations, we link characteristics of the current climate to long-term historical climate simulation. This allows us to evaluate the performance of CMIP6 models, and to constrain both local and remote processes mediating the ARLF. Point 3 on the role of clouds and boundary-layer dynamics is treated separately from this process-oriented constraint. Our model-based results in Fig. 2 are thereby linked to a deeper study of these perspectives in large-eddy simulations.

We note that this work aims to provide insights in different perspectives on the ALRF and AA. We bring together a variety of contributions from a large research consortium, and ultimately seek to find synergy among them.

In Section 2 we firstly elaborate on how AA and the Arctic LRF is calculated from climate model diagnostics and radiative kernels, and how to facilitate a constraint based upon this. Secondly, the different observational data sets and individual methods are described. Section 3 evaluates the performance of the two CMIP6 model subsets to simulate processes governing the Arctic LRF, based on the observations introduced in Section 2. In the discussion, we further explain the differences between both model subsets and link our results to the historical climate simulations, which is equivalent to our constraint. Our final conclusions revisit the key hypotheses 1–3.

## 2   Methods

To address the objectives of this study, we evaluate the performance of a subset of CMIP6 models with a wide range of observables in different parts of the Arctic. From CMIP6, we use historical simulations with the best estimates of the transient climate forcings during 1850–2014 (Eyring et al., 2016). In this study, we focus on the period 1951–2014. For our analyses, we use the entire data set of available CMIP6 data and compute ensemble means over all realizations per model. This way, each

model carries equal weight in the inter-model distribution, and we further exclude the chance of choosing one model realization that deviates substantially from the entire population. By taking the average of model realizations over the past decades, we average out the effect of internal climate variability, and isolate the response to external forcing. However, the observations represent a single climate trajectory and thus combine both the effect of internal variability and response to external forcing. We therefore discuss our main results in the context of internal variability (see also Section 2.9).

While monthly-mean data is available for all CMIP6 models used in this study, only a few models provide all diagnostics necessary for comparing data at higher time resolutions. Therefore, we define three different model data sets at different time resolutions: monthly (all 31 models), daily, and 6-hourly. We specify the models that provide all necessary diagnostics per time-resolution group in Table 1. The model data for each of these time-resolution groups are further broken down into a respective subset that simulates an either weak, or strong historical AA and ALRF (CMIP6/w, or CMIP6/s, respectively). For 150   CMIP6/w, and CMIP6/s subsets we group together the three models with lowest, and highest simulated AA, respectively (see Table 1 for details). Thereby, we largely focus on climate models at the edge of the inter-model range to ensure a clear signal, and allow for an attribution to either weak or strong historical AA/ALRF projection. We do not perform a "classic" emergent constraint that seeks strong statistical relationships between aspects of past or future climate simulations, and the observable present. The idea is to group together models below, and above the observed magnitude of AA to classify weak, and strong 155   historical AA/ALRF simulations, respectively, and to falsify either of them that are less represented by observations.

   We further use observational estimates to calculate AA, and to interpret the simulated model range with respect to observations. The "best" estimate of AA is derived from the AA averages from NASA's Goddard Institute for Space Studies Surface Temperature version 4 (GISTEMP), the Berkeley Earth temperature dataset (BEST), the Met Office Hadley Centre/-Climatic Research Unit version 5.0.1.0 (HadCRUT5), NOAA's Merged Land-Ocean Surface Temperature Analysis (MLOST), 160   and ERA5 reanalysis.

   In each comparison step, we use a specific observational data set to evaluate the performance of the respective CMIP6/w and CMIP6/s subset, and to constrain one key process mediating the ALRF. The location of the observational sites are summarised in Fig. 3. The model-to-observation (or model-to-reanalysis) comparisons include the following:

- We compare temperature inversion strengths measured during the Multidisciplinary drifting Observatory for the Study
of Arctic Climate (MOSAiC; Shupe et al., 2022) to corresponding CMIP6 data. The color coding in Fig. 3 shows the drift of the research vessel *RV Polarstern* during MOSAiC from October 2019 to October 2020. More information is given in Section 2.2.

- Complimentary to MOSAiC, we further use inversion data from long-term radiosonde observations at the Atmospheric Radiation Measurement (ARM) site at Utqiaģvik, Alaska, USA (see Section 2.3 for details).

- We further analyse measurements of temperature profiles by dropsondes released from research aircraft during measurement campaigns in the Fram Strait (gray box in Fig. 3). More information about the campaign data is provided in Section 2.4.

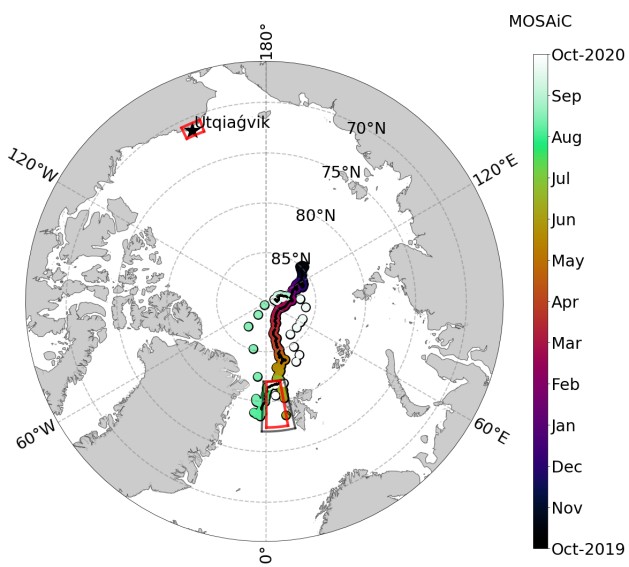

**Figure 3.** The Arctic region north of 66° N summarising all domains considered for comparing observations and reanalyses to CMIP6. Color coding represents the location of the *RV Polarstern* drift as a function of time from October 2019 to October 2020 during the MOSAiC expedition. Black dots in the track represent location and time of the observational data set used in this study (Sec. 2.2). The ARM site at Utqiaġvik, NSA is marked by a star (Sec. 2.3). The dropsonde domain is the enclosed area in the Fram Strait marked by the grey trapezoids (Sec. 2.4). The regions around Utqiaġvik and the Fram Strait discussed in Sec. 2.5 are marked by red trapezes. The entire area north of 66° N is used for deriving Pan-Arctic averages of Arctic LRF and AA (Sec. 2.1). Additionally, we consider the net energy transport across the Arctic boundary, and the long-wave radiation budget at the top-of-the-atmosphere within this area (Section 2.6 and 2.7, respectively).

– In the context of remote controls on the ALRF, we relate the depth of the Arctic warming at the observational sites at Utqiaġvik and the Fram Strait (enclosed by red sectors in Fig. 3) with preferred large-scale atmospheric circulation regimes over these regions. Further information is given in Section 2.5.

– To broaden the perspective of advective controls, we derive the Pan-Arctic poleward energy transport across the Arctic boundary at 66° N latitude, which encloses the entire area illustrated in Fig. 3. The methodology is further specified in Section 2.6.

– Finally, the LRF as a positive radiative feedback in the Arctic locally enhances the Greenhouse effect. Therefore, we relate its strength to changes in the long-wave radiation budget at the top of the atmosphere (TOA). Again, we consider the area north of 66° N to derive Pan-Arctic averages. Details are described in Section 2.7.

– In an outlook perspective, to augment the observational data sets derived during MOSAiC, we further conducted daily Large-Eddy Simulations (LES) for the whole MOSAiC drift (Section 2.8). These simulations aid a discussion of pro-

cesses at turbulence- and cloud-resolving scales, as they are largely underrepresented in the literature covering the Arctic LRF.

## 2.1 The Arctic amplification and the Arctic LRF in CMIP6

To facilitate constraints on the past AA and ALRF, we first calculate a model-specific AA and ALRF value from monthly-mean temperature fields for all 31 climate models considered in this study. We define the degree of AA by subtracting the global-mean near-surface air temperature change, $\Delta T_{\mathrm{s}}$, from the respective Arctic-mean. Arctic mean values account for the averaged area north of $66°$ N. We chose this metric over defining AA the ratio of global-mean and Arctic warming as in the period of interest, some model realizations show a global-mean warming that is close to zero. Therefore, the ratio estimator brings the risk to arbitrarily inflate the model spread (Hind et al., 2016; Davy et al., 2018).

The LRF arises from tropospheric warming that is vertically non-uniform. The change in temperature profiles is calculated for the averaged period of the last 30 years of the historical simulations (1985–2014) compared to 1951–1980. By choosing the time period, we cover the modern era of Arctic amplification that has been identified from the second half of the 20th century, continuing into the 21st century (Davy et al., 2018 and references therein.) The feedback is derived from pre-computed radiative kernels which give the change in TOA radiation balance due to a perturbation of the temperature by $1\,\mathrm{K}$. We considered radiative kernels from the CAM5 (Pendergrass, 2017), GFDL AM2 (Feldl et al., 2017), ERA-Interim (Huang et al., 2017), and HadGEM3-GA7 (Smith et al., 2020) climate models. The model-specific LRF is derived as LRF average from each of the kernels. The corresponding kernel-average ALRF values are given in Table 1, with inter-kernel standard deviations as uncertainty ranges. We want to stress that the inter-model relationship between AA and ALRF is only slightly affected by the choice of kernel, with correlation coefficients of $r = 0.89, 0.90, 0.91, 0.92$ for HadGEM3-GA7, CAM5, ERA-Interim, GFDL AM2, respectively. Therefore, our classification as either CMIP6/w or CMIP6/s models is not sensitive to the choice of kernel.

The feedback parameter $\lambda$ is defined as

$$\lambda = \frac{\partial R}{\partial X}\frac{\Delta X}{\Delta T_{\mathrm{s}}}, \tag{1}$$

with $\frac{\partial R}{\partial X}$ representing the kernel. $\Delta X$ gives the change in temperature profile that deviates from $\Delta T_{\mathrm{s}}$. The LRF is calculated by applying Eq. (1) and integrating over the troposphere. We derive the tropopause following Soden and Held (2006), by defining the $100\,\mathrm{hPa}$ pressure level as tropopause at the equator, and using a linear slope (according to geographical latitude) down to $300\,\mathrm{hPa}$ at the poles.

The feedback parameter $\lambda$ has units of $\mathrm{Wm^{-2}K^{-1}}$. We redefine the feedback parameter as a warming contribution to $\Delta T_{\mathrm{s}}$, by using the local energy budget following several prior studies (Lu and Cai, 2009; Crook et al., 2011; Feldl and Roe, 2013; Taylor et al., 2013; Pithan and Mauritsen, 2014; Goosse et al., 2018; Hahn et al., 2021):

$$0 = F + \left(\sum_i \lambda_i + \lambda_{\mathrm{P}}\right)\Delta T_{\mathrm{s}} + \Delta\mathrm{OHU} + \Delta\mathrm{AHT} = F + \left(\sum_i \lambda_i + \overline{\lambda_{\mathrm{P}}} + \lambda'_{\mathrm{P}}\right)\Delta T_{\mathrm{s}} + \Delta\mathrm{OHU} + \Delta\mathrm{AHT}. \tag{2}$$

The local energy budget in Eq. (2) describes the energetic contributions of the radiative forcing $F$, the feedbacks ($\lambda_i \Delta T_{\mathrm{s}}$) and the Planck response ($\lambda_{\mathrm{P}}\Delta T_{\mathrm{s}}$), as well as the ocean heat uptake ($\Delta\mathrm{OHU}$), and the anomalous atmospheric heat transport

convergence ($\Delta$AHT). The second step splits the Planck feedback in its global-mean value, $\overline{\lambda_\mathrm{P}}$, and the spatially resolved deviation from it, $\lambda'_\mathrm{P} = \lambda_\mathrm{P} - \overline{\lambda_\mathrm{P}}$. Therefore, we can derive the warming contributions to $\Delta T_s$ of forcings and feedbacks by dividing each term in Eq. (2) by the global-mean Planck feedback ($\overline{\lambda_\mathrm{P}}$):

$$\Delta T_\mathrm{s} = -\frac{\left(F + \Delta \mathrm{OHU} + \Delta \mathrm{AHT}\right)}{\overline{\lambda_\mathrm{P}}} - \frac{\left(\lambda'_\mathrm{P} + \sum_i \lambda_i\right)\Delta T_\mathrm{s}}{\overline{\lambda_\mathrm{P}}}. \tag{3}$$

In that form, each of the individual contributions on the right-hand side add up to the full change in $T_s$. In our study however, we only consider the contribution of the LRF. This approach offers the advantage of emphasising the physical connection between local radiative fluxes and the local temperature change by climate feedbacks (Feldl and Roe, 2013).

## 2.2 Temperature inversions from radio soundings during MOSAiC

During the MOSAiC expedition between October 2019 and October 2020, the *RV Polarstern* drifted within the central Arctic
sea ice. During the expedition, among others, vast atmospheric measurements were carried out (Shupe et al., 2022). In this study, we analyse thermodynamic profiles from Vaisala RS41-SGP radiosondes that were launched at least four times per day (Maturilli et al., 2021). In order to estimate inversion strengths from the soundings, we additionally employ concurrent 2-m temperature (T2m) measurements from the nearby MOSAiC ice camp (Cox et al., 2021), since the soundings were launched from the ship's helicopter deck approximately 10-m above the ice, thereby missing the lowermost meters of the atmospheric
column. In addition, using the T2m tower data reduces the impact of the ship on the near-surface temperature. We derive the inversion strength as the difference between the temperature-profile maximum (Tmax; between the surface and $250\,\mathrm{hPa}$) and T2m. Each model's vertical resolution is thereby maintained without interpolating the profiles to a common pressure coordinate.

The temporal resolution for the inversion data follows the frequency of radiosonde launches during the MOSAiC expedition
(approximately every 6 hours). For the model-to-observation comparison, we consider 6-hourly temperature diagnostics for the period 2010–2014 that were co-located to MOSAiC in space and time. Since the climate models are free-running coupled models, it is not essential to use the exact years of 2019–2020, but instead the correct time (i.e., time of day and season), and spatial location are co-located. Nevertheless, we justify the model-to-observation comparison by testing the similarity of model time series for historical output data 2000–2014, and the highest emission scenario data (SSP585) as upper boundary of the
range of scenarios, for those models that provide 6-hourly diagnostics for both simulations (not shown). Our analysis shows that the SSP585 time series consistently lies within the inter-annual range of 2000–2014 historical data, and for most of the year, within the range of inter-annual standard deviation.

The model output data is chosen corresponding to which time step and grid-box midpoint is closest to each individual MOSAiC radiosonde launch time. Essentially, the model data "follow" the MOSAiC track in space and time of the year. We
equally derive the temperature inversion in the model data as the difference between Tmax and T2m.

Note that there are no inversion data available at MOSAiC between 9 May 2020 to 10 June 2020, and 29 July 2020 to 25 August 2020 when the ship was in transit through the sea ice. Figure 3 shows the entire drift of *RV Polarstern* with time

attribution according to the radiosonde launches as color coding. The black dots following the drift depict the locations where observational data was available for our study (limited by the availability of T2m tower data).

### 2.3 Temperature inversions from radio soundings at Utqiaġvik (NSA)

The ARM program organised by the U.S. Department Of Energy (DOE) provides a long-term record of atmospheric observations from permanent and mobile measurement sites around the world (Mather and Voyles, 2013). One ARM site that is particularly relevant for Arctic studies is the North Slope of Alaska (NSA) in Utqiaġvik, Alaska, USA. With a geographical location of 71.23° N and 156.61° W, the NSA site is one of the most important sources for long-term western Arctic atmospheric observations, which makes it ideal for climate studies.

For this study we use atmospheric temperature profiles from radiosonde launches performed at the NSA site. The so-called interpolated sounding (INTERPSONDE) Value Added Product is obtained after linearly interpolating the atmospheric state variables from consecutive soundings into a fixed 2-D time-height grid. The grid's temporal resolution is 1 min. The vertical resolutions varies with altitude, ranging from 20 m in the lowest 3.5 km, to 50 m between 3.5–5 km, to 100 m between 5–7 km, and to 200 m between 7–20 km altitude, respectively. It is important to mention that the input for the INTERPSONDE-product comprises only data from quality controlled soundings and precipitable water vapour estimated from microwave radiometer measurements, and it does not incorporate ancillary observations from surface or tower meteorological observations. The INTERPSONDE-product's fixed 2-D grid facilitates the comparison with weather and climate models. Radiosonde data for the NSA site are available since April 2002, with a varying 2 to 4 launches per day (Jensen et al., 1998). The overlapping time period available for analysis constrains the data set to the range of April 2003 to December 2014.

Once CMIP6 model output and NSA radiosonde data are processed to be comparable, we estimate the inversion strength as in the MOSAiC comparison, as the difference between Tmax and T2m, and at 6-hourly time resolution. We only take into account inversion strengths larger than 0.5 K to dismiss cases that resemble adiabatic lapse-rates.

### 2.4 Temperature profiles from dropsondes in the Fram Strait

The relationship between the ALRF and the strength of sea ice retreat motivates the assessment of temperature profiles above both sea-ice and open-ocean surfaces, as well as their differences. For this purpose, measurements of dropsondes released from research aircrafts in the Fram Strait are analysed. The dropsondes deliver atmospheric profiles for altitudes below the launch location. The limited flight altitude of the employed research aircrafts constrains the maximum altitude of the resulting temperature profiles to about 3 km. Since the measurements presented here are available only for March, we restrict the model-observation-comparison to this month. However, the thermodynamic conditions are similar as compared to the extended winter season, DJFM.

In total, 52 dropsondes are analysed, which were launched mainly in an area between 77–82° N and 2° W–13° E (see Fig. 3) during the following three campaigns: eight sondes during the Radiation and Eddy Flux Experiment (REFLEX, performed in March 1993; Lüpkes and Schlünzen, 1996), 22 sondes during the Spring Time Atmospheric Boundary Layer Experiment (STABLE, performed in March 2013; Lüpkes et al., 2021), and 22 sondes during the Airborne measurements of radiative and

turbulent FLUXes of energy and momentum in the Arctic boundary layer campaign (AFLUX, performed in March/April 2019; Becker et al., 2020).

For surface type classification, the sea ice concentration at the dropsonde launch location was obtained from satellite observations (Kern et al. (2020) for REFLEX, and Melsheimer and Spreen (2019) for STABLE and AFLUX). If the sea ice concentration was below 15 %, a profile is considered to represent conditions over open ocean, while a sea ice concentration above 85 % corresponds to sea-ice covered ocean. Thereby, we exclude data from 6 dropsondes that were launched over the marginal sea ice zone (15–85 %) in this analysis, which is designed to obtain a clear signal for the difference between sea ice and open ocean.

As for MOSAiC and NSA, the model-to-observation comparison applies data with 6-hourly time resolution for 2010–2014 in the model output. Similar to the observations, the temperature profiles from the models were grouped into open ocean and sea ice conditions based on the model sea ice concentration at the respective grid cell. The location of the sea ice edge varies significantly among the models. To reduce the impact of the different distances to the sea ice edge on the thermodynamic profile, grid points with a distance of more than 250 km to the 50 % isoline of sea ice concentration are excluded from the analysis.

## 2.5 The role of advective heating

Not only can atmospheric stability and sea-ice loss mediate the thermodynamic structure of the atmosphere, but also remote influences. Here, we link the vertical structure of Arctic warming to large-scale atmospheric circulation regimes over the regions of the Fram Strait and Utqiaġvik (marked in Fig. 3). Again, the years of 1951–1980 are chosen as the reference period, and 1985–2014 represent the present-day climate state.

We identify preferred atmospheric circulation regimes in reanalysis data by analyzing daily mean sea-level pressure (SLP) anomaly fields over the North-Atlantic-Eurasian region (30–90° N, 90° W–90° E) and over the North-Pacific region (30–90° N, 90° E–90° W) separately for the extended winter season (DJFM). For the reanalysis data, the 5th generation reanalysis of the European Centre for Medium-Range Weather Forecasts is employed (ERA5; Hersbach et al., 2020). We follow the approach described in Crasemann et al. (2017) and determine the ERA5-based circulation regimes as non-Gaussian structures in a reduced state space (Dawson and Palmer, 2015). In more detail, the analysis comprises the following steps: 1) The dimensionality of the data set is reduced by an Empirical Orthogonal Function (EOF) analysis. The subsequent analysis is performed in the reduced state space spanned by the five leading EOFs (Dawson and Palmer, 2015), which explain about 57.5% of the variance of the SLP anomaly fields over the North-Atlantic-Eurasian region and 54.8% of the variance over the North-Pacific region. The leading EOFs resemble well-known teleconnection patterns such as the North-Atlantic Oscillation, East Atlantic pattern, Pacific/North American pattern, West Pacific pattern. The coordinates in the reduced state space are provided by the corresponding non-normalized Principal Component (PC) time series. We have proven the robustness of the identified regimes with an analysis in the state space, spanned by the 10 leading EOFs. 2) A k-means-clustering has been performed in the reduced state space where the number of clusters $k$ has been set to $k = 5$ following Crasemann et al. (2017). These clusters are interpreted

as preferred circulation regimes and each time step of the data set has been assigned to one of the clusters. The clusters are characterized by SLP anomaly fields, reconstructed from the 5-dimensional coordinate vectors of the cluster centroids.

For the analysis of the CMIP6 data we apply a projection approach described in Fabiano et al. (2021) where the state space spanned by the ERA5-EOFs serves as the reference state space for the CMIP6 simulations. The coordinates for each simulation are provided by projecting the SLP anomaly data onto the reference state space, obtaining five Pseudo-PCs, for each model simulation. Based on these Pseudo PCs, each day of the respective model simulation is assigned to the closest centroid of the five ERA5 reference clusters. The advantage of this approach is the consistent definition of the atmospheric circulation regimes.

A bootstrap test similar to Crasemann et al. (2017) was used to test for changes in the relative frequency of occurrence of the regimes between the reference and the present-day period. A significant change in the frequency of occurrence was detected at the 95% level if no more than 5% of 10,000 bootstrap replicates of the time series describing the occurrence of the regimes showed a greater difference than the change in the frequency of occurrence of the original occurrence time series.

In order to relate the occurrence probability of each circulation regime i ($P_i$, i $= 1, ..., 5$) to the vertical structure of the warming at the observational sites, we applied a Multi-Nomial Logistic Regression (MNLR) approach. This approach was used by e.g. Detring et al. (2021) to study recent trends in blocking probabilities, but it is also suitable for the multi-class problem of describing $P_i$ in dependence of some covariates. The basic idea of MLNR is to describe the log-odds (defined as the logarithm of the chance of observing a distinct regime with respect to a predefined base-line regime) as a linear combination of the covariates. For our analysis, the covariates comprise the 2-m temperature (T2m), the mid-tropospheric temperature at $500\,\mathrm{hPa}$ (T500), and time. T2m and T500 are averaged values over the region around the respective measurement site.

Finally, the relationship between the occurrence probability of each circulation regime and the warming structure is expressed as a 2-dimensional PDF dependent on T2m and T500 changes. We henceforward refer to an increase in T2m, and T500 with time as bottom-heavy warming, and top-heavy warming, respectively. We constrain the remote influence of advective heating on the ALRF by a model-to-reanalysis comparison, using ERA5 and CMIP6 models with daily output data as specified in Table 1.

We ultimately seek to establish a link between changes in large-scale circulation patterns that mediate vertically non-uniform warming structures, and the local magnitude of the LRF in the Arctic. In a second step, we extent this method and focus on the Pan-Arctic atmospheric transport in the current climate and its connection to both past AA and ALRF.

## 2.6 Pan-Arctic atmospheric energy transport

To derive the Pan-Arctic atmospheric transport in the present-day climate state, we make use of the large-scale and long-term Arctic atmospheric energy budget (AEB) equation. Following previous works (e.g., Nakamura and Oort, 1988; Trenberth, 1997; Serreze et al., 2007), we can describe the energy budget of any atmospheric column that extends from the surface to the TOA as

$$\frac{\partial E_a}{\partial t} = R_a + Q_H - \nabla \cdot \boldsymbol{F}_a, \tag{4}$$

comprising the tendency in energy storage $\frac{\partial E_a}{\partial t}$, the net atmospheric radiation budget $R_a$, the sum of turbulent heat fluxes at the surface $Q_H$, and the convergence of the horizontal atmospheric energy transport $-\nabla \cdot \boldsymbol{F}_a$. The radiation budget $R_a$ is derived as the sum of the net downward radiative flux at the TOA and the upward radiative flux at the surface in both long and short-wave frequencies, respectively. The net turbulent heat flux at the surface is composed of both sensible and latent heating. The AEB equation in the form of Eq. (4) is a simplification and does not account for factors like the conversion between liquid water and precipitating ice. However, the residual that arises from these terms is shown to be small in the long-term and annual mean Arctic AEB, just as the storage tendency under the steady-state assumption (Serreze et al., 2007; Linke and Quaas, 2022). The main components that define the long-term and large-scale Arctic AEB are therefore the atmospheric radiation budget, the net surface turbulent heat flux, and the transport term. We apply the same approach as in Linke and Quaas (2022), and derive the horizontal convergence of energy transport indirectly, i.e., as residual of the AEB equation. From the indirect method of using the energy budget, we do not distinguish either contributions of dry static energy and latent heat transport.

For our constraint, we compare the transport convergence (positive = net atmospheric transport into the polar cap) at present-day climate state (2000–2014) in a model-to-reanalysis comparison. Due to the larger amount of model data available in the subset with monthly resolution (Table 1), we further calculate inter-model correlation coefficients for the entire collection of models.

For determining statistical significance in our analysis, a bootstrap method based on 10000 samples was used. Correlation coefficients with a two-tailed p-value less than 0.05 were considered statistically significant.

## 2.7 Pan-Arctic outgoing long-wave radiation at the TOA

Our last constraint for past ALRF and AA exploits changes in the outgoing long-wave radiation at TOA ($OLR_{TOA}$) during past decades. Theoretically, both the magnitude of AA and ALRF reflect in the $OLR_{TOA}$ and its evolution with time.

We compare CMIP6 models against two data records from satellite observations (all-sky broadband radiation fluxes), and ERA5 reanalyses, respectively. The first satellite data record is derived from the Advanced Very High Resolution Radiometer (AVHRR) afternoon orbit (PM) sensors aboard the Polar Operational Environmental Satellite (POES) missions (Stengel et al., 2020). The data record covers the period 1982–2016 and was funded by the European Space Agency (ESA) as part of the ESA Climate Change Initiative (CCI) program. Although the morning (AM) sensor series was available, it was found that only the PM series has the radiometric stability needed for trend studies (Lelli et al., 2023). The second satellite record is produced by NOAA/NCEI from the High Resolution Infrared Radiation Sounder (HIRS) instruments on board the NOAA and MetOp satellites (Zhang et al., 2021). It provides the OLR flux at TOA since 1979, thereby offering observations over more than 40 years. We average all three data records to derive a "best combined" (BEST COMB) estimate of OLR data.

Changes in the $OLR_{TOA}$ data records are derived as linear trends (least squares polynomial fit) in the period 1983–2014. Thereby, we do not cover the entire period of historical CMIP6 simulations ongoing from 1951 to address the change (like in Section 2.5), but instead use the overlap period between the beginning of the satellite record (1983; starting with the full year) and the end of the historical CMIP6 simulations (2014).

## 2.8 The role of advection, clouds and entrainment in large-eddy simulations (LES)

While the MOSAiC observational data sets (partly addressed in Section 2.2) are unprecedented in their coverage of the low level thermal structure in the central Arctic, various aspects that play a key role in the ALRF were not continuously sampled. These include processes such as turbulent entrainment driven by cloud top cooling across shallow liquid layers. To augment the observational data set, we conduct daily LES for the whole MOSAiC drift, at turbulence- and cloud-resolving scales. The four-dimensional output of these simulations is used as a virtual laboratory to address how small-scale boundary layer processes affect the thermal structure of the lower atmosphere within a heat budget framework. Covering the full MOSAiC drift with such simulations is a significant computational effort and goes far beyond the more common application of LES for short, single case studies. The added value of this effort is that it allows for bridging the gap between small-scale, fast-acting atmospheric boundary-layer processes and long-term means at climate time-scales (Neggers et al., 2012).

The daily LES experiments for MOSAiC were conducted with the DALES code (Heus et al., 2010). The simulated domain is Eulerian, situated around the location of the *RV Polarstern*. The domain size is $0.8 \times 0.8 \times 12 \, \text{km}^3$, discretised at a grid-size of $8 \times 8 \times 288$. The horizontal grid-spacing is $100 \times 100 \, \text{m}^2$, while for the vertical dimension a telescopic grid is used featuring a vertical resolution of $10 \, \text{m}$ across the lowest $2 \, \text{km}$. A previous LES study using such micro-grid LES experiments (Neggers et al., 2019) showed that at this resolution and domain size, the turbulent entrainment flux is sufficiently resolved. We thus achieve an optimal balance between computational efficiency and spatial resolution to serve our research goals. Subgrid transport is represented using a turbulent kinetic energy (TKE) scheme, while cloud microphysics are represented using the bulk double moment mixed-phase scheme as described by Seifert and Beheng (2006), applied to five hydrometeor species. While the Cloud Condensation Nuclei (CCN) concentration is prognostic, affected by processes such as advection, diffusion and microphysics, the concentration of Ice Nucleating Particles (INP) is constant. The radiation is interactive with the model state, as are the surface turbulent fluxes.

The experiments are initialized with the 11 UTC radiosonde profile, interpolated onto the LES grid. Observed CCN (Koontz and Uin, 2016) and INP (Creamean, 2019) concentrations at the surface are used to initialize the associated profiles. The lower boundary condition consists of a prescribed observed skin temperature of sea ice (Reynolds and Riihimaki, 2019) and open water, combined through the observed sea ice fraction. The impact of processes larger than the domain size is represented through prescribed forcings for momentum, temperature and water vapour, derived from ERA5 following the method described by Van Laar et al. (2019) and Neggers et al. (2019). Profiles for horizontal advection tendencies are prescribed, and applied homogeneously in the grid. Vertical advection relies on a prescribed profile of large scale vertical motion, acting on the model state. Composite forcing is applied, meaning that it is time-constant and consists of profiles time averaged over the first 11 hours of each day at the *RV Polarstern* location. As a result, the simulation can equilibrate after spin up. Nudging is applied above the thermal inversion that marks boundary layer top, with nudging linearly increasing in intensity across a $1 \, \text{km}$ deep transition layer towards full nudging above, at a relaxation timescale of $1800 \, \text{s}$. Below the inversion, no nudging is applied, leaving the turbulence and clouds free to evolve.

## 2.9 Internal variability

In each of the above described methods, we compare observations/reanalyses to co-located model data of ensemble means. By taking the average of all model realizations over the past decades, we average out the effect of internal variability, and isolate the response to external forcing. As such, the differences between CMIP6/w and CMIP6/s subsets can be attributed to external forcing. The observations/reanalyses however, represent a single climate trajectory and thus combine both the effect of internal variability and response to external forcing. When comparing the observations/reanalyses to the model output (from CMIP6/w and CMIP6/s, respectively) it is thus important to discuss if constraining the simulated parameters is justified when accounting for internal climate variability in the model data. We therefore examine whether the differences between observations/reanalyses and model simulations can be explained by internal variability within each subset. In particular, we compute the differences (observations/reanalyses minus CMIP/w and CMIP6/s, respectively) and compare that difference to the respective range of model realizations which is attributable to internal variability. This range is calculated by subtracting the ensemble mean from each realization (to remove the forced response), and then calculating the central 95 % range per model subset (e.g. England et al., 2021). If the observation/reanalysis − model difference lies without that range, it cannot with confidence be explained by internal variability, which justifies falsifying the specific model subset based on the constraint.

## 3 Results

In the following, we revisit all aspects of the current climate system introduced in Section 2 in the scope of a model-to-observation/reanalysis comparison. We first present the basis on which our constraints are built in Section 3.1: The large spread among CMIP6 models in simulating the magnitude of historical ALRF and AA, and their inter-model relationship. Following that, we compare each individual observational/reanalysis data set to co-located weak and strong-AA/ALRF model output data, and falsify either one or the other. We consider first the local, surface-near Arctic processes in Section 3.2.1 to 3.2.3 by comparing temperature data from radio soundings and dropsondes to the co-located model data. We then transition from local to remote processes, that can further affect the higher troposphere in Section 3.3.1 and 3.3.2. Section 3.4 serves as a summarising result of the individual aspects, by considering changes in the long-wave radiation budget at the TOA. Section 3.5 gives an outlook on the role of clouds and boundary layer dynamics.

## 3.1 Relation between AA and ALRF in CMIP6 models

Firstly, the scatter plot in Fig. 1b shows the spread in AA among climate models, which linearly relates to their spread in ALRF ($R = 0.87$). Thereby, models with a higher magnitude of AA have a stronger positive ALRF. The "best" observational estimate (OBS) indicates that more models over-predict the simulated value of AA and consequently ALRF, whereas less models underestimate the OBS. However, the OBS AA magnitude of 0.67 K is close to the center of the simulated model range (0.68 K AA). Thereby, our classification of CMIP6/w and CMIP6/s subsets by grouping together the three models with lowest

and highest simulated AA, respectively, ensures that the subset averages lie below and above the OBS AA value. This justifies the categorization as weak or strong-AA models with respect to observations.

Secondly, Fig. 1a shows a clear distinction of the CMIP6/w and CMIP6/s model subsets from the multi-model mean, and the results from these models naturally fall outside the ensemble standard deviation. In addition, the stronger contribution to AA from the CMIP6/s collection arises from the combination of both a more negative, and positive radiative feedback in the tropics, and the Arctic, respectively. This, however, does not necessarily relate to the inter-model spread in global warming: The linear correlation coefficient between global warming and AA/ALRF is 0.51/0.45, respectively.

All models used in this study are specified in Table 1, including the model-specific AA and ALRF, corresponding to Fig. 1b. Again, we use different models for representing CMIP6/w and CMIP6/s model subsets, depending on the time resolution. The model usage is specified in Table 1 by a superscript in the model acronym. Note that individual time-resolution groups always apply for the same models. For instance, Section 2.2–2.4 compare model and observational data at 6-hourly resolution. Thereby, the CMIP6/w subset includes data from the SAM0-UNICON, MPI-ESM-1-2-HAM and AWI-ESM-1-1-LR, and

CMIP6/s from CNRM-ESM2-1, CNRM-CM6-1-HR and IPSL-CM6A-LR model, respectively.

### 3.2   Local aspects: Temperature inversion and temperature profiles

In a first step, we evaluate the ability of CMIP6 models to simulate the omnipresent surface-based temperature inversion, just as temperature profiles in the Arctic. We compare inversion data derived from radiosondes and weather stations during the MOSAiC expedition and at Utqiaġvik (NSA), just as temperature profiling from several dropsondes in the Fram strait, to

co-located model data with 6-hourly time resolution.

### 3.2.1   Temperature inversions during MOSAiC

Figure 4 shows the comparison between inversion measurements during the MOSAiC expedition, and co-located simulated inversion data for the CMIP6/w and CMIP6/s subsets, respectively.

     The time series in Fig. 4a depicts, on average, a stronger inversion for the CMIP6/w subset during boreal fall (ON) and

extended winter (DJFM). In turn, during spring (AMJ), the CMIP6/s subset shows slightly stronger inversions, on average. For summer (JAS), both model groups have similar inversion strengths. The differences between both model subsets are most noteworthy during ONDJFM.

     We propose the following to explain the relation between present-day inversion and historical LRF in the Arctic: The stronger inversion in CMIP6/w in the present-day period during ONDJFM is consistent with the negative relationship between ALRF

and the change in inversion strength among climate models (Boeke et al., 2021): A stronger Arctic LRF corresponds to more bottom-heavy warming in the past, i.e. a stronger depletion of the surface temperature inversion. This explains why CMIP6/s models end up having a weaker inversion in the present-day period.

     During extended winter, the CMIP6/w models are in better agreement with the observations compared to the CMIP6/s subset (compare to box plots in Fig. 4b). During ON, the observations lie in between both sub-groups, but still closer to the CMIP6/w

average. During AMJ, both subsets of the models tend to overestimate the inversion strength from the observations. However,

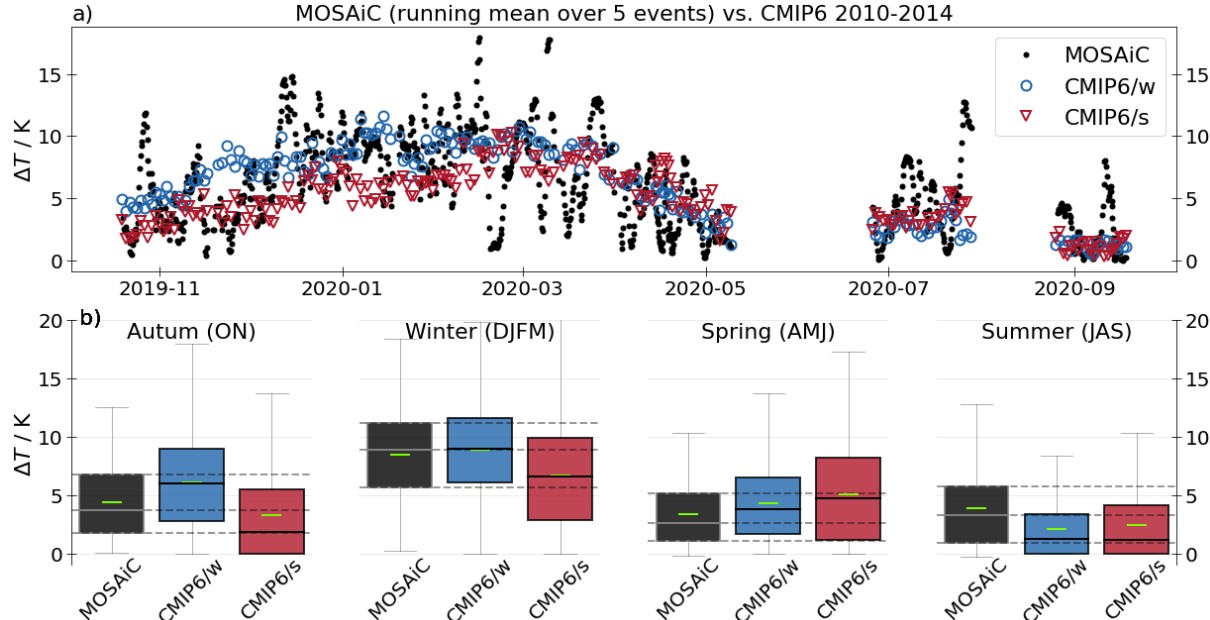

**Figure 4.** a) Inversion strengths $\Delta T$ obtained from radio soundings during the MOSAiC expedition, and for the model subsets CMIP6/w and CMIP6/s, respectively. The time series of radio soundings during MOSAiC is given as rolling average over 10 launches. b) Seasonal inversion strengths $\Delta T$ as box plots, corresponding to panel a. Boxes show the 25 to 75-percentile of the data, whiskers the 5 to 95-percentiles, grey horizontal lines in the boxes the median values, and green horizontal lines the mean values. MOSAiC data was collected during Oct 2019 to Oct 2020, and is compared to co-located 6-hourly model data in the period of 2010–2014. Details on the data processing are given in Section 2.2.

the CMIP6/w subset is slightly closer to the observations. During JAS, in turn, both subsets show inversions that are too weak in comparison to MOSAiC observations. However, severe data gaps during spring and summer make the interpretation somewhat less reliable.

It is noteworthy that during MOSAiC, a number of anomalous events were detected, e.g. extreme cases of warm, moist air transported from the northern North Atlantic or northwestern Siberia during late fall until early spring (Rinke et al., 2021). That raises the question if MOSAiC inversion data is an appropriate choice for constraining climate models. Rinke et al. (2021) compare the near-surface meteorological conditions during MOSAiC to the context of the recent climatology, and show that for the full time series, the temperature at 2 m, and 850 hPa lies mostly within the record, even during storms and moisture intrusion events. We thereby expect that the temperature inversion is representative for climatological averages. Another line of evidence is that the winter-time inversion during MOSAiC is similar to the winter-time inversion during the SHEBA (Surface Heat Budget of the Arctic) campaign (approx. 8 K in the averaged DJF temperature profile; Stramler et al., 2011).

In summary, from the presented comparative time series, we particularly emphasise the results presented during ONDJFM: The *RV Polarstern* drifted within the central Arctic, mostly north of 85 ° during that time (Figure 3). CMIP6/w models simulated

a stronger present-day inversion than CMIP6/s, and are closer to the observed distribution during the MOSAiC expedition during (ON)DJFM. The model subsets during DJFM are clearly distinguishable, also by the range of individual models: The average inversion strength from those three models in the CMIP6/w, and CMIP6/s subset lie within 7.6–10.6 K and 5.8–6.9 K, respectively. During ON, the CMIP6/w, and CMIP6/s subset results lie within 4.6–5.8 K, and 1.8–3.5 K, respectively. Primarily during DJFM, the MOSAiC inversion average (8.49 K) is most attributable to the range of CMIP6/w models. We further elaborate on the statistical representativeness of the results during DJFM by explicitly showing the three distributions of CMIP6/w, CMIP6/s, and MOSAiC inversion data as histogram in Fig. A1. It is noteworthy that the model subsets show a shift of distribution towards lower inversion values for CMIP6/s models. We further perform a two-sample Kolmogorov-Smirnov test to compare the similarity of CMIP6/w and CMIP6/s distributions (not shown explicitly). The test indicates that both model subsets show a significantly different distribution, and that this difference is largest during DJFM (also large during ON) compared to spring and summer. In addition, the highest correspondence between MOSAiC and model simulations is seen during DJFM, which is supported by Fig. A1.

Regarding the role of internal variability, we note that our conclusion that CMIP6/w models more realistically represent the MOSAiC inversion data is based on the comparison to ensemble means. However, individual CMIP6/s realizations might still be consistent with the observed inversion. Fig. B1a indicates that this is not the case: The bar plots show the averaged inversion during DJFM, observed and simulated (ensemble averages; corresponding to Fig. 4b). The gray bars further indicate the residuals after subtracting CMIP6/w and CMIP6/s averages (externally forced response) from the observations. The error bars account for internal variability of the respective model subset (Section 2.9). The internal-variability range of CMIP6/s models does not fully cover the MOSAiC − CMIP6/s difference. The difference can therefore not be explained with confidence by internal variability of the CMIP6/s ensemble, as it is the case for the smaller MOSAiC − CMIP6/w difference. This justifies our conclusion that CMIP6/s models systematically underestimate the inversion during DJFM. The same applies for a similar comparison during ON and AMJ (not shown).

### 3.2.2  Temperature inversions at Utqiaġvik (NSA)

The regular radiosonde observations at the Utqiaġvik site are complementary to the MOSAiC analysis in that they provide long-term statistics, albeit at one site, and are representative of a different geographical (coastal) region in the Arctic. We present our results in comparison with the measurements conducted during MOSAiC. Correspondingly, the co-located model data cover the period of 2010–2014 and apply the same CMIP6 models as defined in Table 1.

During both ON and DJFM, CMIP6/w models, on average, show a stronger inversion compared to the CMIP6/s subset, and vice versa during AMJ, which is consistent with the findings for the MOSAiC data in Fig. 4. This agreement with the findings from MOSAiC suggests the same explanation also holds true for this longer-term analysis.

The comparison of the observed inversion data with the ones from models shows that the CMIP6/w model subset lies closer to the observations in ON. For the winter case, it is somewhat less clear than for the MOSAiC comparison: The observations lie in between CMIP6/w and CMIP6/s with regard to the 25 and 75-percentile of the data. The average inversion at NSA is closer to the subset average of CMIP6/w, but the median is closer to CMIP6/s. We expect this differences compared to the MOSAiC

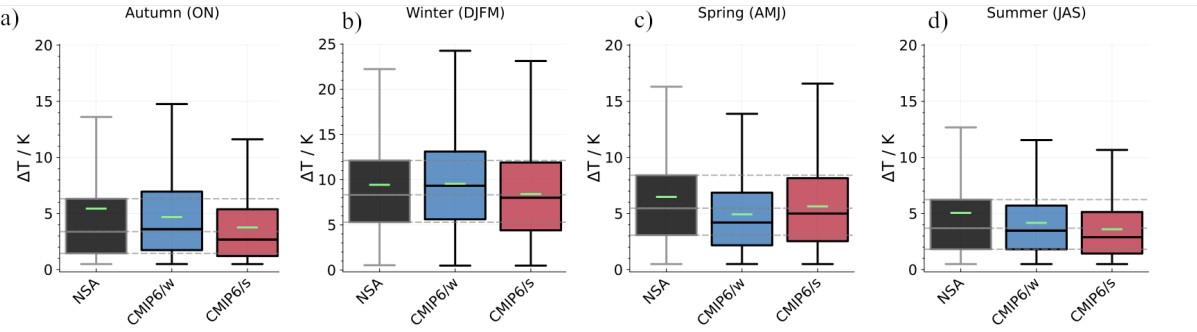

**Figure 5.** Seasonal inversion strengths $\Delta T$ as box plots, obtained from radio soundings at the North Slope of Alaska (NSA) in Utqiaġvik, Alaska, USA, and from the model subsets CMIP6/w and CMIP6/s, respectively. The box plots correspond to Fig. 4b showing the seasonal distribution of $\Delta T$ from radio soundings during the MOSAiC expedition. NSA data was collected during 2003–2014, and is compared to co-located 6-hourly model data in the period of 2010–2014. Details on the data processing are given in Section 2.3.

analysis to be linked to the vicinity of ocean at the NSA site. In the following section, dropsonde measurements show that CMIP6/w models overestimate atmospheric stability over ocean, but CMIP6/s models simulate less stable conditions during
the month of March. This would explain that the inversion strengths derived at NSA lies somewhat in between both subsets. In addition, the model data for both subsets is less clearly distinguishable as compared to the MOSAiC sampling.

In spring, the CMIP6/w models underestimate the inversion strength as compared to the observations, while CMIP6/s models fits the observations better. This is in contrast to our MOSAiC results, which suggest that both model groups overestimate the inversion strength at this time of the year. However, due to large data gaps for MOSAiC during this season, caution should
be taken while interpreting the results. During JAS, the inversion strength is underestimated in all models, a result that is in agreement with the MOSAiC data.

In summary, we find links between the model-to-observation comparison for the MOSAiC expedition, and at the NSA site. In particular, we find that both analyses transfer from a period where CMIP6/w model have stronger inversions (ONDJFM) to a period where CMIP6/s models simulate more stable conditions (AMJ). Where the observations are deviating from the model
average inversion strength (MOSAiC; DJFM, and NSA; ON), we find that the stronger inversions as simulated by CMIP6/w models more realistically represent the observations. In addition, during ONDJFM, the MOSAiC − CMIP6 (ensemble mean) difference lies within the range of internal variability for CMIP6/w, but not for CMIP6/s (not shown), as for MOSAiC.

### 3.2.3 Temperature profiles in the Fram Strait

In order to assess the mediating effect of the surface type (open ocean or sea ice) on the temperature profile, we make use of
540 dropsonde profiles launched from aircraft. Again, this analysis is complimentary to the results from the MOSAiC and NSA data comparison, embedded in the context of local influences on the Arctic LRF. We thereby apply the same models, but only include data during the end of extended winter (March), as discussed before in Section 2.4. The comparison with co-located

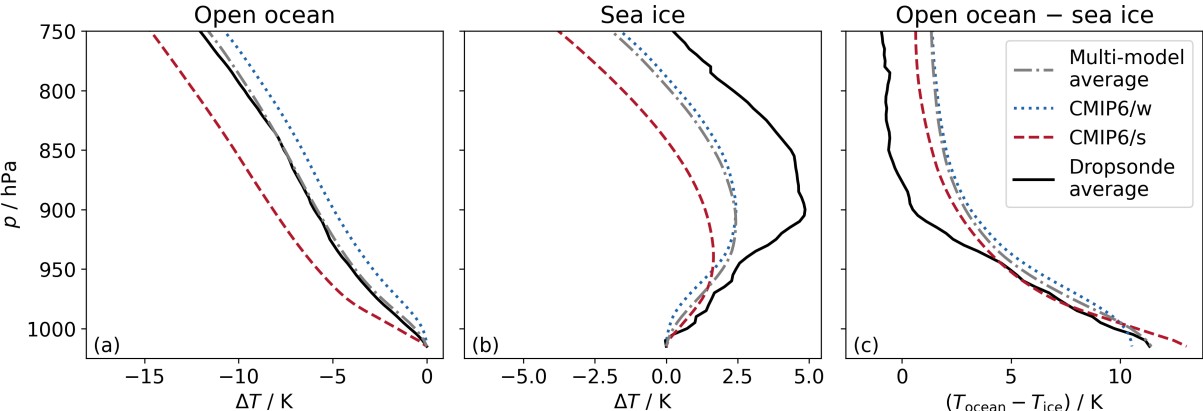

**Figure 6.** Average profiles of temperature normalized with the temperature at $1015\,\mathrm{hPa}$ ($\Delta T$) over a) open ocean, and b) sea ice, respectively. $\Delta T$ is obtained from dropsonde launches during aircraft campaigns in the Fram Strait, and the model subsets CMIP6/w and CMIP6/s, just as the model average, respectively. Seasonally, our results are restricted to the month of March. Panel c shows the difference between the temperature profiles over open ocean and sea ice. Dropsonde data was collected during three flight campaigns in 1993, 2013, 2019, and is compared to co-located 6-hourly model data in the period of 2010–2014. Details on the data processing are given in Section 2.4.

CMIP6/w and CMIP6/s model subsets is shown in Fig. 6. Panel a and b show the temperature profiles derived from observations and models over open ocean and sea ice, respectively, which are normalised to the temperature at $1015\,\mathrm{hPa}$. Note that due to a
lack of open ocean data in the CNRM-ESM2-1 model (number 25 in Table 1) domain, CMIP6/s only comprises two models.

     The mean temperature profiles derived from both models and observations show an almost linear temperature decrease over open ocean, as also expected climatologically (Fig. 2a and b). In contrast, the profile over sea ice shows a near-surface temperature inversion for both observations and model data (again, in agreement with the climatological analysis in Fig. 2). Over ocean, the CMIP6/s subset shows slightly less stable conditions than the CMIP6/w data. Similarly, the CMIP6/w subset
simulates a stronger inversion (on average $4.35\,\mathrm{K}$), compared to the CMIP6/s data (on average $3.55\,\mathrm{K}$) over sea ice. The inversion strength is derived as in previous sections, as difference between Tmax and T2m. The stronger simulated stability in present-day temperature profiles as projected by the CMIP6/w subset is in agreement with previous results from MOSAiC in the central Arctic, and the NSA site located near the coast during fall and winter. Note that the difference in stability between both subsets weakens when including campaign data from April (not shown). We attribute this to the fact that during AMJ, both
MOSAiC and NSA show a transition to CMIP6/s models simulating stronger present-day inversions compared to CMIP6/w. This likely leads to less differences between the subsets in the dropsonde data through overlapping signals between March and April. Overall, both model subsets underestimate the inversion strength compared to the observations over sea ice. However, over both open ocean and sea ice, the CMIP6/w subset is closer to the observations, albeit being rather consistent with the multi-model average.

To analyse the impact of sea ice retreat on the temperature profile, Fig. 6c shows the difference in profiles between open-ocean and sea-ice areas. Close to the surface, the temperature difference between ocean and sea ice is larger for the CMIP6/s

subset (on average 13.0 K) compared to CMIP6/w (on average 10.5 K). This is mostly due to higher near-surface air temperatures over ocean in the CMIP6/s subset (not shown). However, above 1000 hPa the situation reverses, with a larger surface-type temperature difference for CMIP6/w models. Comparing to the observations, the warming expected through sea ice retreat is slightly better depicted by the CMIP6/w models very close to the surface. However, in higher layers, CMIP6/s models simulate a slightly more realistic temperature difference between profiles over ocean and sea ice (albeit the difference between models subsets is small).

We conclude that in the context of simulated stability over sea ice, the dropsonde results representing the month of March are in agreement with the inversion data obtained from the central Arctic during MOSAiC, and at the coast of the NSA site during DJFM. This concerns the stronger simulated stability by CMIP6/w models, and their closer match with observations during DJFM over sea ice, as shown by the MOSAiC-observation-to-model comparison. We further show that when switching from sea ice to open ocean, the CMIP6/s models generate a stronger increase in the near-surface air temperature than the CMIP6/w models, but less warming in the higher troposphere. Both implies a stronger contribution to a positive LRF embedded in the processes driving the Arctic LRF: bottom-heavy warming and muted top-heavy warming. Our data however is temporally limited and accounts solely for the month of March.

### 3.3    Remote aspects: Atmospheric energy transport

Up to this point, we have presented results that concern the local and surface-near Arctic temperature structure and their link to the simulated past AA and Arctic LRF. We now focus on the impact of remote controls, by firstly extending our results shown in Fig. 6c, i.e. the evolution of bottom-heavy and top-heavy warming, and their potential to mediate the vertical warming structure in a model-to-reanalysis comparison.

### 3.3.1    The role of advective heating

In this analysis on advective bottom- and top-heavy warming, we focus on the same area of the Fram Strait as in the previous section, and further include the observational site of Utqiaġvik (Section 3.2.2). Bottom-heavy warming conceptually addresses the key feature of the Arctic LRF, i.e. the stronger warming of near-surface air masses compared to aloft. Top-heavy warming on the contrary describes the concept of stronger warming in higher layers of the tropospheric column, as compared to the surface. To address these vertically non-uniform warming structures, we analyse changes in the occurrence of those transport pathways that are related to either BHW or THW during extended winter (DJFM), during the time period of interest (1985–2014 with respect to 1951–1980). Thereby, we link vertically non-uniform warming structures to circulation patterns, and further explore the potential impact on the local LRF at site. To evaluate the performance of CMIP6 models, we compare CMIP6/w and CMIP6/s model subsets to ERA5 data. The transport pathways are characterised in terms of preferred atmospheric circulation regimes and the warming profiles are described in terms of T2m and T500 anomalies (Section 2.5 for details).

The transport pathways over the Fram Strait region (0°–10° E, 77.4°–82° N; see Fig. 3) are characterised by the five distinct circulation regimes over the North Atlantic-Eurasian region (e.g., Crasemann et al., 2017), namely the Scandinavian/Ural blocking regime (SCAN/Ural), the negative phase of North Atlantic Oscillation (NAO-), the dipole pattern regime (DIPOL),

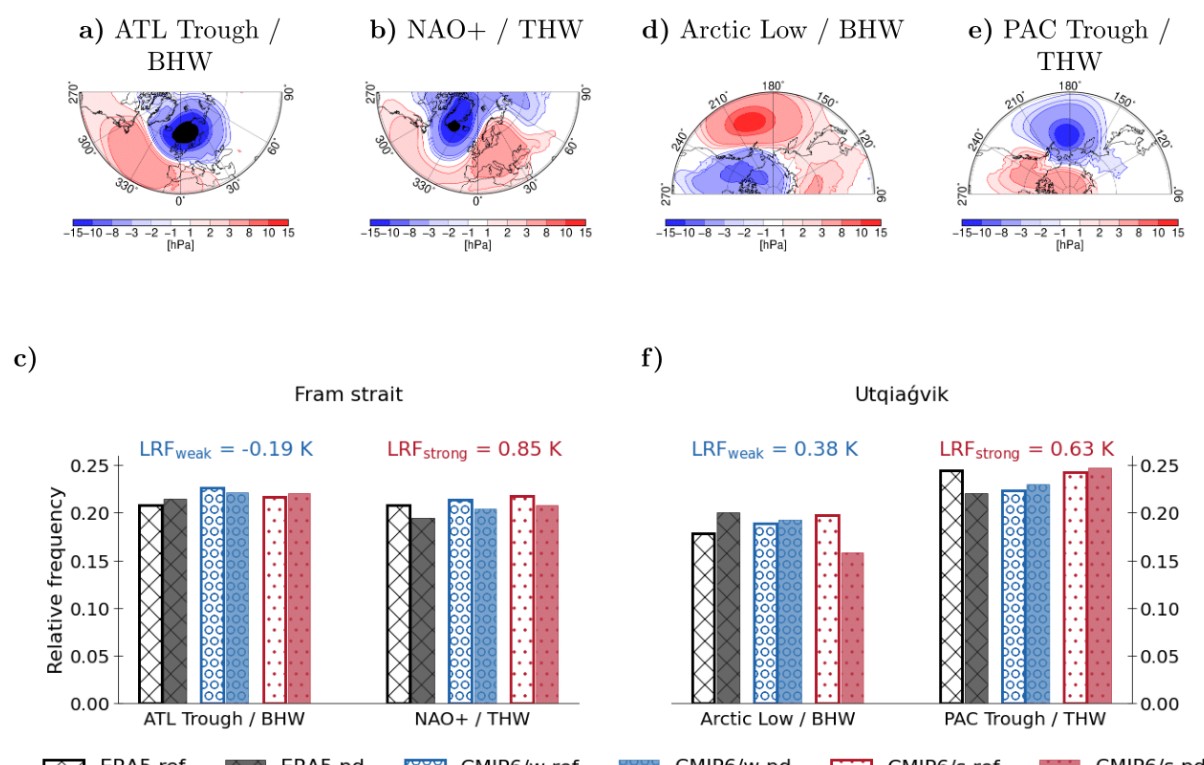

**Figure 7.** Changes in the relative frequency of circulation regimes associated with either bottom-heavy warming (BHW) or top-heavy warming (THW) for ERA5, and the model subsets CMIP6/w and CMIP6/s, respectively. The left side of the plot refers to the North-Atlantic-Eurasian region (a, b, c), the right side to the North-Pacific region (d, e, f). Upper rows show the circulation regimes, lower rows their frequency of occurrence for reference (ref) and present-day (pd) period, respectively. Seasonally, we focus on the extended winter period DJFM. North-Atlantic-Eurasian region: SLP anomaly patterns of the two circulation regimes which are related to a) strong BHW (Atlantic Trough; ATL Trough) and b) strong THW (NAO+), based on ERA5 daily mean SLP data for 1979–2020. c) Changes in the relative frequency of occurrence between the reference and the present-day period of the respective regimes over the Fram Strait. North Pacific region: d) and e) as in a) and b): SLP anomaly patterns of two circulation regimes which are related to d) strong BHW (Arctic Low) and e) strong THW (Pacific Trough; PAC Trough), based on ERA5 daily mean SLP data for 1979-2020. f) as in to c): Changes in the relative frequency between reference period and present-day period of the respective regimes over Utqiaġvik. The reference and present-day period in ERA5 (CMIP6) is 1979–1999 (1951–1980) and 2000–2020 (1985–2014), respectively. The values above panels c and f give the local LRF for CMIP6/w (LRF$_{weak}$) and CMIP6/s (LRF$_{strong}$) over both domains, respectively. We use daily output data for both ERA5 and CMIP6 in this analysis. Details on the data processing are given in Section 2.5

the Atlantic trough regime (ATL Trough), and the positive phase of North Atlantic Oscillation (NAO+). The application of the MNLR approach described in Sec. 2.5 reveals a high occurrence probability of the ATL Trough regime for BHW over the Fram Strait for ERA5 (Fig. 7a) as well as for the climate models (not shown). The occurrence of strong THW over the Fram Strait

is associated with a high probability of the NAO+ circulation regime (Fig. 7b). For ERA5, Fig. 7c shows that the ATL Trough regime (associated with BHW) occurs more frequently, and the NAO+ regime (associated with THW) less frequently in the present-day period compared to the reference. Although non-significant, both of these changes imply a potentially positive feedback contribution of advection to the Arctic LRF. For the CMIP6/w models, both the ATL Trough and the NAO+ regime occur less frequently in the present-day period with the implication of counteracting effects on the local LRF by advection. On the other hand, for the CMIP6/s models, the ATL Trough regime occurrence increases and the NAO+ regime occurrence decreases in the present-day period. We suggest that the differences in the sign of occurrence changes in the ATL Trough / BHW regime are related to the differences in the strength of ALRF, comparing the two model subsets over the Fram Strait region.

The transport pathways over the Utqiaġvik region (200°–205.9° E, 70.6°–71.8° N; see Fig. 3) are characterised by five distinct circulation regimes over the North Pacific region (e.g., Amini and Straus, 2019), namely the Pacific Trough (PAC Trough), the Arctic High, the Pacific wave train, the Arctic low and the Alaskan ridge regime. By applying the MNLR approach, a high occurrence probability of the Arctic low regime for BHW for ERA5 (Fig. 7d), as well as for the climate models (not shown) has been detected. The occurrence of THW over the Utqiaġvik region is related to a high probability of the PAC Trough regime (Fig. 7e). For ERA5, Fig. 7f shows that the Arctic low regime (associated with BHW) occurs more frequently, and the PAC Trough regime (associated with THW) less frequently in the present-day period. Again, both of these changes in the remote influences, (which at Utqiaġvik, have passed the bootstrap significance test described in Sec. 2.5), can positively contribute to the Arctic LRF. For the CMIP6/w models, both the occurrence of the Arctic low regime and the PAC Trough regime increases slightly in the present-day period, with the implication of counteracting effects of advection on the local LRF. For the CMIP6/s models, the Arctic low regime occurrence decreases and the PAC Trough regime occurrence increases in the present-day period, which potentially contributes to a weakening of the positive LRF through advection over the Utqiaġvik region.

In summary, at both sites of the Fram Strait and Utqiaġvik, CMIP6/w and CMIP6/s model subsets differ from each other in terms of their changes in relative frequency of BHW regimes in the present-day period 1985–2014 with respect to 1951–1980: In the Fram-Strait domain, CMIP/w models show a decrease in the relative frequency in BHW, while CMIP6/s models shows an increase. Both have less THW in the present-day period. Thereby, we suggest a negative LRF contribution to CMIP6/w, and a positive LRF contribution to CMIP6/s models through the influence of advective BHW in the Fram Strait, respectively. At Utqiaġvik, the situation is reversed. CMIP/w models show an increase in the relative frequency in BHW, while CMIP/s models show a decrease. Both have more THW in the present-day period. Thereby, we suggest a positive LRF contribution to CMIP6/w, and a negative LRF contribution to CMIP6/s models through the influence of advective BHW at Utqiaġvik, respectively. We link these differences between the model subsets at both locations to the magnitude of the co-located LRF (values given in Fig. 7c and f) later on in the discussion.

In terms of their similarities with ERA5 results, the changes in advective BHW/THW show that the CMIP6/s models have a closer resemblance to ERA5 over the Fram Strait. At Utqiaġvik, the ERA5 data however, show an opposite tendency in the evolution of BHW and THW in comparison with CMIP6/s models. For the CMIP6/w models, only the increase in BHW is consistent with ERA5, albeit less pronounced in the models. Note that by applying the bootstrap test, we determine significant

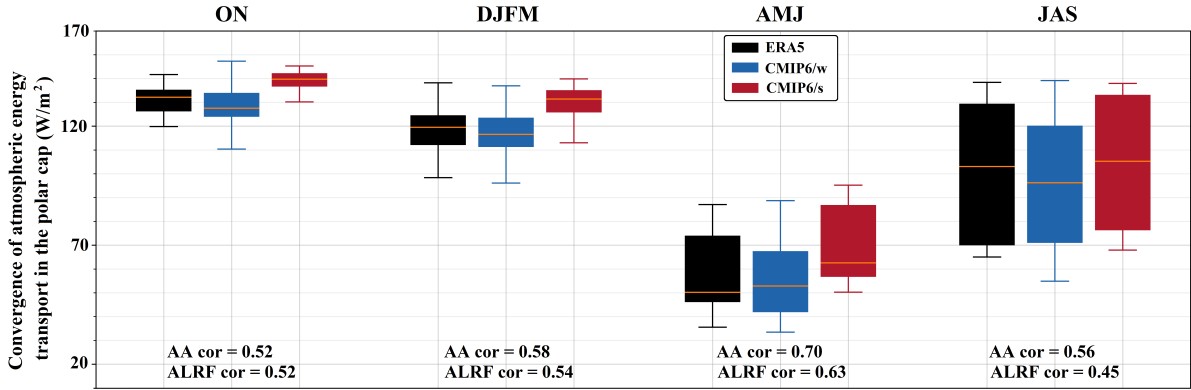

**Figure 8.** Seasonal Pan-Arctic atmospheric energy transport convergence north of 66° N, obtained from ERA5, and the model subsets CMIP6/w and CMIP6/s, respectively, during 2000–2014. The seasonal inter-model correlation coefficients between the seasonal mean of transport term and historical AA/ALRF (including all models in Table 1) are given in the lower part of each panel. We use monthly output data for both ERA5 and CMIP6 in this analysis. Details on the derivation of atmospheric energy transport convergence are given in Section 2.6.

changes at the 95% level only at the site of Utqiaġvik (for changes in BHW/THW in ERA5, and in BHW for CMIP6/s models). Albeit the attribution of model simulated results to reanalyses remains somewhat illusive, we argue that the differences in the change in occurrence of BHW, and their link to the local LRF at site motivates a more extensive investigation of the link between large-scale circulation regimes that impact the evolution of local vertical warming structure, and local differences in Arctic temperature feedbacks.

### 3.3.2 Pan-Arctic atmospheric energy transport

In the second step of considering remote controls on the Arctic LRF, we extend the perspective of energy transport to a broader view. Figure 8 depicts the total poleward atmospheric energy transport convergence within the Arctic boundary during each season for ERA5, CMIP6/w, and CMIP6/s subsets, respectively. The present-day transport accounts for the averaged period of 2000–2014 for both CMIP6 and reanalysis data. Firstly, it is shown that deriving the atmospheric energy transport convergence as the residual of the energy budget equation (see Section 2.6 for details) gives a realistic approximation of the seasonal cycle of the Arctic energy transport: During late fall and winter, the atmospheric energy transport convergence into the polar cap shows a seasonal maximum due to the absence of solar irradiance. The Arctic atmosphere is in an approximate balance between long-wave radiative cooling and the advection of energy from lower latitudes (Cronin and Jansen, 2016). During spring and early summer, the long-wave cooling intensifies due to higher atmospheric temperatures, but the incoming solar radiation adds a heat source to the atmosphere, which leads to a decrease in the seasonal atmospheric transport into the polar cap (e.g., Trenberth, 1997; Serreze et al., 2007; Linke and Quaas, 2022).

From the differences between CMIP6/w and CMIP6/s simulations, it is shown that the present-day poleward transport convergence is stronger for the CMIP6/s subset as compared to CMIP6/w models. This is true for each season, and furthermore,

the difference appears to be systematic across the entire model ensemble: The transport convergence and AA/ALRF are positively correlated across all models, which is shown by the inter-model correlation coefficients in the lower part of Fig. 8. This correlation is particularly strong during AMJ, with AA cor $= 0.70$, and ALRF cor $= 0.63$, but it is always above 0.5 (except for ALRF cor during JAS). In addition, the seasonal inter-model mean correlation of the transport term with both AA and ALRF is statically significant throughout the year, using the bootstrapped method with a 0.95 confidence level (see Section 2.6 for details). We therefore conclude that the model differences in simulated atmospheric energy transport convergence are systematic in that stronger-AA models show a stronger present-day transport, and vice versa for weaker-AA models.

To evaluate which of the model subsets more realistically projects the atmospheric transport into the polar cap, we compare both CMIP6/w and CMIP6/s simulations to ERA5 results. The box plots in Figure 8 show that the reanalyses is closer to the transport as simulated by CMIP6/w models during ONDJFM. During AMJ, we find no clear difference in the ability of both subsets to resemble the reanalyses estimated transport convergence. During JAS, the box plot in Figure 8 implies that CMIP6/s models more realistically simulated the atmospheric transport convergence into the polar cap.

In summary, we find that during each season, models with weaker (stronger) present-day poleward transport convergence simulate a smaller (larger) past Arctic LRF and AA, with statistical significance. For the model-to-reanalyses comparison, we find that during ONDJFM, CMIP6/w models with lower present-day transport convergence more realistically resemble ERA5 results. During JAS, CMIP6/s models more realistically resemble the transport estimate of ERA5, but the model differences, and thereby their attribution to reanalyses is less clear compared to ONDJFM.

We further show in Fig.B1b that during DJFM, the difference between the average transport in ERA5, and CMIP6/w (ensemble mean) can be mostly explained by internal variability within the CMIP6/w subset, albeit not fully. However, due to the positive correlation between AA/ALRF and transport convergence ($r = 0.58$ / $r = 0.54$, respectively), the difference is reduced when choosing e.g., the three next highest models for CMIP6/w, which is then covered by internal variability (not shown). We thereby conclude that CMIP6/s models more likely overestimate the energy transport convergence.

Lastly, we find similarities in previous model-to-observation comparisons of local vertical temperature structures: Our results in Section 3.2 have shown that during fall (ON) and extended winter (DJFM), but primarily during DJFM, CMIP6/w models more realistically resemble observations of the surface-based temperature inversion over sea ice. We find now that also in the representation of processes that further concern the free troposphere (energy transport from lower latitudes), CMIP6/w models are closer to reanalyses, primarily during DJFM.

## 3.4 Pan-Arctic outgoing long-wave radiation at the TOA

As introduced earlier, the global LRF builds on either limited atmospheric cooling in the long-wave spectrum (as in the Arctic), or an intensification of this process (i.e. a reduced Greenhouse effect like in the tropics). Thereby, the LRF (amongst other feedbacks and forcings) mediates changes in TOA energy budget in the long-wave radiation spectrum. In our final step, we investigate changes in the outgoing long-wave radiation at the TOA during past decades. Within the scope of the TOA energy budget, we seek to constrain the overall LRF in the Arctic by a model-to-satellite/reanalysis data comparison, covering the period of 1983–2014 (from the beginning of the full-year satellite/reanalysis records to the end of the CMIP6 simulations).

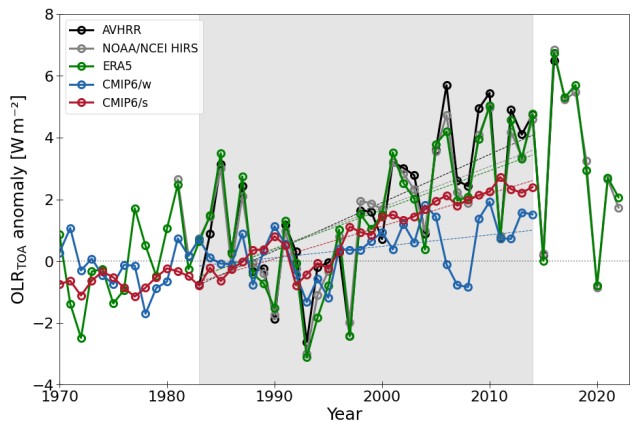

**Figure 9.** Time series of $OLR_{TOA}$ anomaly with respect to 1983–1997 for AHVRR, NOAA/NCEI HIRS and ERA5 climate data records (averaged to provide a best combined record of OLR data; BEST COMB), just as the model subsets CMIP6/w and CMIP6/s, respectively. Trends are derived for both satellite observations/reanalyses, and model subsets as linear fit for the data overlap period (1983–2014; shaded area). The trends are $0.138 \pm 0.017$, $0.037 \pm 0.010$, and $0.106 \pm 0.009 \, \mathrm{Wm^{-2}yr^{-1}}$ for BEST COMB, CMIP6/w, and CMIP6/s, respectively. The uncertainty ranges account for the standard deviation of trends using the bootstrap method of Lelli et al. (2023). The fluxes are averaged over the Arctic area north of $66°$ N, and account for the extended winter period DJFM. For the comparison, we use the collection of monthly-mean model diagnostics. Details on the data processing are given in Section 2.7.

Due to previous links found between inter-mediate conclusions in Sections 3.2.1–3.3.2, we focus on the most relevant winter season DJFM.

Figure 9 depicts an overall increases in $OLR_{TOA}$ within the period of interest which is consistent with atmospheric warming. The CMIP6/s subset shows a stronger increase in $OLR_{TOA}$ compared to CMIP6/w which coincides with a notoriously stronger warming in the CMIP6/s simulations (not shown). However, we cannot detect a signal of the stronger past Arctic LRF in the CMIP6/s subset here, as this would imply less $OLR_{TOA}$, compared to the CMIP6/w simulations (again, the LRF increases the local Greenhouse effect in the Arctic). Thereby, the stronger warming effect overshadows the feedback signal.

Looking at the anomaly in OLR with respect to 1983–1997 (Figure 9) the BEST COMB trend (average of AVHRR, NOAA/NCEI HIRS, and ERA5 records) shows a stronger increase in $OLR_{TOA}$ ($0.138 \pm 0.017 \, \mathrm{Wm^{-2}yr^{-1}}$) compared to both CMIP6/w and CMIP6/s subsets (with an increase of $0.037 \pm 0.010 \, \mathrm{Wm^{-2}yr^{-1}}$ and $0.106 \pm 0.009 \, \mathrm{Wm^{-2}yr^{-1}}$, respectively). Overall, the CMIP6/s subset is closer to the BEST COMB trend during 1983–2014, but still underestimates the increase in $OLR_{TOA}$. In a TOA perspective, both model subsets are under-representing the change in increasing OLR with advanced global warming. This links to a general lack in the ability of climate models to project the magnitude of Arctic climate change during the most recent decades (discussed late on).

Including the aspect of internal climate variability, Fig. B1c indicates a large spread across individual realizations for both model subsets (error bars). The conclusion that CMIP6/s simulations more realistically represent the observed OLR trends is supported by the smaller difference between BEST COMB and CMIP6/s, compared to CMIP6/w. This difference lies within the

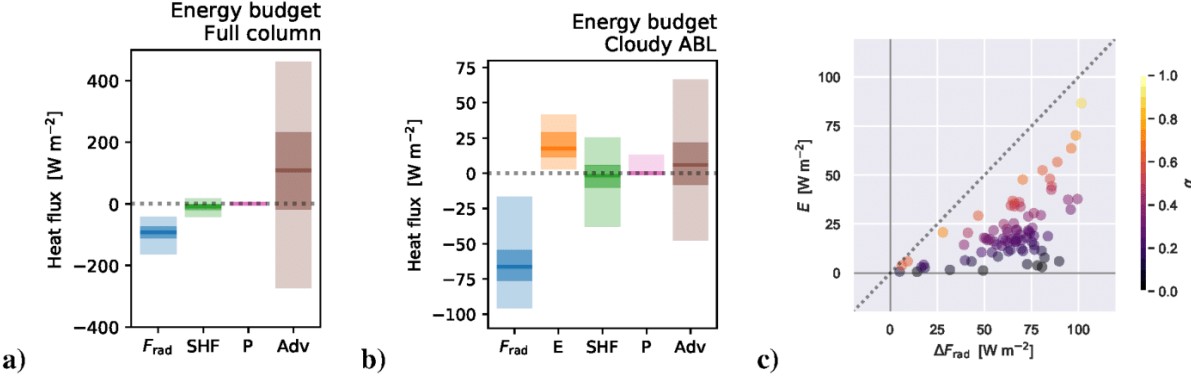

**Figure 10.** LES results for the MOSAiC drift. a) Drift-average heat budget of the full atmospheric column. b) Conditionally averaged ABL heat budget for all days with a non-zero liquid water path (83 out of 300 simulated days). Budget terms include net radiative heating ($F_{rad}$), top entrainment heat flux ($E$), surface sensible heat flux (SHF), surface precipitation in any form ($P$), and vertically integrated large-scale advection (Adv). Each term is shown as a distribution, with the median (thick line), interquartile range (dark shaded) and 9-95 percentile range (light shaded). c) Scatter plot of daily average long-wave cooling across the liquid cloud layer versus the entrainment heat flux at the atmospheric boundary layer inversion. Coloring represents the entrainment efficiency $\alpha$, as defined in the text. Details on the data processing are given in Section 2.8.

range of internal variability as simulated by CMIP6/s. On the other hand, the BEST-COMB − CMIP6/w difference cannot be fully explained by the range of internal variability simulated by CMIP6/w models, which justifies the conclusion that CMIP6/w
models systematically underestimate the OLR trend.

Up to this point, we compared key features of the Arctic LRF and AA in the current climate to co-located model simulations with both weak and strong simulated ALRF and AA in the past. Our model-to-observation/reanalysis comparisons covered the key aspects of Arctic temperature inversions, sea ice retreat, local advection and Pan-Arctic atmospheric energy transport, and the link between AA/ALRF and changes in the TOA long-wave radiation budget with warming. On the other hand, processes
at turbulence- and cloud-resolving scales are largely underrepresented in the literature covering the Arctic LRF. In our final step, we show the potential of these processes to impact the evolution of the Arctic temperature profile. We thereby link to our results in Fig. 2 which briefly motivated the role of clouds in the evolution of the Arctic LRF in a purely CMIP6-based analysis. The next section deepens this analysis in the scope of a local energy budget perspective in large-eddy simulations covering the MOSAiC drift. We treat this section separately from the constraint approach, and thereby drop the comparison to CMIP6
simulations in an outlook perspective.

### 3.5 The role of advection, clouds and entrainment in large-eddy simulations (LES)

To gain insight in the role of clouds and boundary-layer dynamics, we now investigate the Arctic energy budget in more detail, using output data from year-long LES covering the MOSAiC drift. The results are shown in Fig. 10.

Averaged over all 300 daily simulations we find that the full column heat budget is approximately in radiative-advective equilibrium (RAE; Fig. 10a). This RAE has been previously introduced as basic-state model for representing the high-latitude atmosphere (Cronin and Jansen, 2016). Even if only a certain number of weather situations were sampled during the drift, and even if the scatter in particular in the advective heating remains large, this confirms what is expected for the large-scale energy budget of the ice-covered Arctic: In particular, the surface flux (SHF) and precipitation ($P$; representing net condensation and freezing in the column) are negligible compared to the radiative cooling ($F_{\mathrm{rad}}$) and vertically integrated advective heating (Adv).

The situation is rather different, though, when i) only analysing cases with low level liquid cloud mass, and ii) considering the heat budget for the Atmospheric Boundary Layer (ABL). For the whole drift about 1 out of 3 days features low level liquid clouds. This frequency of occurrence is roughly consistent with the findings of Bennartz et al. (2013), and is an expression of the resilience of mixed phase clouds at high latitudes (Morrison et al., 2012). In contrast to the full column, the ABL heat budget shows an imbalance in which the radiative cooling dominates (Fig. 10b). On average, this leads to a gradual cooling of the ABL in cloudy cases, which likely expresses the ongoing transformation of warm and moist air masses in which these clouds are embedded (Pithan and Mauritsen, 2014; Pithan et al., 2018).

An intriguing result in the context of the LRF is the significant role played by entrainment at the top of the ABL, here defined as the height of the strongest gradient in liquid water potential temperature ($\theta_l$) in the lowest 5 km. In these cloudy cases, in addition to the weak advective heating, the warming of the ABL due to the entrainment flux ($E$) is significant, while sensible and latent heat flux are again negligible as was the case in all-sky conditions (Fig. 10b). The entrainment heating ($E = \epsilon_t \, \Delta\theta_l$) depends on the ABL-top entrainment rate ($\epsilon_t$) and the temperature jump across the inversion ($\Delta\theta_l$), an expression of local (elevated) inversion strength. Entrainment warming can only counteract the radiative cooling partially, an effect which is investigated in further detail in Fig. 10c. The impact on the ABL heat budget is expressed by the entrainment efficiency ($\alpha$), defined as the ratio of entrainment warming to the radiative cooling (Stevens et al., 2005):

$$\alpha = \frac{\epsilon_t \, \Delta\theta_l}{|F_{\mathrm{rad}}|}. \tag{5}$$

We find, on the basis of these year-long LES results for the MOSAiC drift, that for mixed phase Arctic clouds this ratio is about 1/3, implying that entrainment warming is never able to fully balance the radiative cooling. However, it still significantly counteracts the gradual cooling of warm and moist air masses that enter the Arctic system. In this process, the main role of inversion strength is to determine the entrainment warming. As a result, it modulates the transformation of such warm cloudy air masses, to the effect that it keeps them warm for longer. This in turn affects the LRF, in particular in ice-covered areas over which such cloudy air masses travel.

## 4  Discussion

We have presented data from several Arctic-based observations and reanalyses in conjunction with co-located CMIP6 model simulations to constrain various processes-relevant parameters that mediate both Arctic amplification and the Arctic LRF. We

thereby exploit the considerable inter-model spread in simulated AA and ALRF, which are linearly related across CMIP6 models. For the linear relationship between AA and ALRF we show that models with stronger positive ALRF contribute more to AA, both through locally enhancing global warming in the Arctic, and cooling the tropics, which does not necessarily reflect in the inter-model spread of projected global warming.

Our process-oriented constraints attribute observable aspects of the current climate system to co-located CMIP6 simulations from models that project an either weak or strong AA and ALRF in the past. This allows us to establish a link between key aspects of the current climate, and the evolution of AA and ALRF in the past. The magnitude of ALRF and AA for the historical past is defined as the time period of 1951–2014.

For our constraint, we firstly make use of the hypothesis that the AA and the ALRF is related to the lack of boundary-layer mixing in the Arctic. Previous literature, since the earliest global dynamical simulations of climate change, demonstrate that stable stratification is a necessary condition for a positive LRF in the Arctic (Manabe and Wetherald, 1975). This leads to the hypothesis that the ALRF inter-model spread is correlated to the change in inversion strength, associated with bottom-heavy warming (Boeke et al., 2021; Feldl et al., 2020). Another hypothesis suggests that stronger initial stratification produces a more positive feedback (Boeke et al., 2021; Lauer et al., 2020), without there being a consensus among scientists. Here, we look at present-day surface-based temperature inversion data in the scope of a model-to-observation comparison. We use two data sets of radiosondes launched during the MOSAiC expedition in the central Arctic, and from the permanent ARM site at Utqiaġvik (NSA). Using dropsonde observations from research aircraft during several boreal-springtime campaigns, we quantify the contrast in temperature profiles over sea ice and open ocean in the Fram Strait. We aim to constrain the impact of sea ice retreat, which is widely considered as a strong source of bottom-heavy warming.

In spite of their spatio-temporal differences in data acquisition, we find distinct similarities in the individual comparisons of MOSAiC, NSA, and dropsonde data to CMIP6: Our results confirm that during fall and extended winter (ONDJFM), models that simulate a weaker AA/ALRF in the past, have stronger inversions over sea ice in the present. We argue that during fall and winter, the key feature of the positive Arctic LRF, bottom-heavy warming, has led to a stronger depletion of the surface-based temperature inversion in the models with stronger AA/ALRF since 1951–1980. Based on the CMIP6 comparison to dropsonde data, we show that even though the CMIP6/s models show weaker present-day inversions (consistent with MOSAiC and NSA), sea ice melt remains an important process to mediate a stronger bottom-heavy warming, and by extension LRF in future scenarios, compared to CMIP6/w models. Our model-to-observation comparison suggests an overall more realistic depiction of the vertical temperature structure over sea ice by models with weak simulated AA/ALRF in the past during ONDJFM. On the other hand, the residual between observations and CMIP6/s simulations suggests that these models systematically underestimate the temperature inversion, rather than being a manifestation of simulated internal variability.

We want to emphasise that interpreting local and surface-near processes up to this point relies on a small number of models (see Table 1) that provide the required time resolution for a comparison of inversion data. Therefore, our analysis is limited by data availability. For model comparisons with MOSAiC (just as NSA, and dropsonde) data, we use model averages from models 5, 6, and 10 to derive CMIP6/w, and models 25, 28, and 29 for CMIP6/s simulations, respectively. Particularly the CMIP6/w subset does not represent the lower edge of simulated AA range across models (see Fig. 1b). However, the classification as

weak and strong-AA model subset is still justified by the fact that CMIP6/w, and CMIP6/s AA averages lie below, and above the OBS estimate of past AA, respectively. We further tested the sensitivity of the model-to-observation comparison to the number of models chosen for the CMIP6/w and CMIP6/s classification in this time-resolution group. Therefore, we added the next highest / lowest AA-model to the CMIP6/w and CMIP6/s subset (model 11 and 21), respectively. This addition of models has no qualitative effect on our key conclusions that CMIP6/w models show stronger inversions during ONDJFM, and are overall closer to the observations than CMIP6/s. We further emphasise that, albeit the statistical interpretation due to the lack of data remains vague, the combined comparison of radiosonde data during MOSAiC and at the NSA site, just as dropsonde data from flight campaigns in the Fram strait agree on the main emerging points discussed above.

On the other hand, not only local processes have the potential to mediate the ALRF and AA, but also remote influences like the poleward atmospheric energy transport. Firstly, we consider the impact of advective bottom- and top-heavy warming on the local LRF which are connected to changes in typical circulation regimes. The hypothesis is that an increased frequency of occurrence of weather situations favouring BHW imposes a positive contribution on the LRF. In turn, more frequent events of THW aids long-wave cooling in higher layers, thereby weakening the positive LRF. We focus on the period of extended winter (DJFM) in our analysis, and locally, on the observational sites of Utqiaġvik and the Fram Strait. At the two measuring sites, CMIP6/w and CMIP6/s models differ from each other in terms of their change in relative frequency of BHW regimes ongoing from 1951. We suggest a link between the change in advective BHW and LRF at site: Our data show that in the Fram Strait, the difference in LRF between CMIP6/w (-0.19 K) and CMIP6/s (0.85 K) is larger (1.04 K) compared to Utqiaġvik, where the difference in LRF between CMIP6/w (0.38 K) and CMIP6/s (0.63 K) is smaller (0.25 K) by a factor 4. We relate this to an increase (decrease) in BHW in CMIP6/s (CMIP6/w) in the Fram Strait region, where we hence expect an even bigger spread in local LRF between both model subsets, and a decrease (increase) in BHW in CMIP6/s (CMIP6/w) at the site in Utqiaġvik, where we hence expect a reduction in the spread in local LRF between both model subsets. In short, advective BHW increases the climatological spread between CMIP6/w and CMIP6/s simulated LRF in the Fram Strait, and decreases it at Utqiaġvik. Although we cannot exclude that these differences are mediated by changes in the local factors (e.g., sea ice reduction) between the two simulations, our results hint to a signature of advective influences mediating the spatial pattern of the Arctic LRF. Albeit no conclusive results are found in the attribution of either CMIP6/w or CMIP6/s model simulations to ERA5 results, we want to highlight the potential of linking local differences in the model-projected magnitude of the LRF to changes in vertically non-uniform warming structures that are mediated by changes in the large-scale circulation.

While we present our previous results as a detailed analysis of the vertically resolved temperature change and local feedback aspects, we further extent this perspective to speculate on the coupling between Pan-Arctic atmospheric energy transport (convergence) and past AA/ARLF. Our results show that a stronger present-day transport convergence within the Arctic boundary is systematically related to a stronger annual-mean AA/ALRF in the past. It is useful to consider the energetic framework for explaining the positive relationship between present-day atmospheric transport convergence and AA: We specifically show in an analysis of OLR at the TOA, that CMIP6/s models have a stronger cooling tendency in terms of $OLR_{TOA}$. It is likely that the stronger cooling at the TOA due to more advanced Arctic warming in these simulations requires a larger overall atmospheric transport convergence into the polar cap to balance the radiative cooling, and ensure the local energy budget (Linke and Quaas,

2022). To constraint the remote aspects of the current climate, we show that the CMIP6/w models overall resemble the ERA5 transport term more realistically during ONDJFM. On the other hand, the overestimation of energy transport convergence in the CMIP6/s model subset cannot be explained fully by simulated internal variability.

To finalise the constraint of past AA and ALRF, we compare recent trends of OLR at the TOA, and its magnitude in the current climate to observational estimates. The CMIP6-derived $OLR_{TOA}$ trends of the past 30–40 years underestimate the observations. Recent work of Rantanen et al. (2022) shows that since 1979, the Arctic has warmed more drastically than previously thought, and that CMIP6 models under-represent the warming trend that is depicted by observations. We see a link to these results when exploring the trend in OLR at the TOA since 1983 during DJFM: Both CMIP6/w and CMIP6/s models show lower trends in the increasing $OLR_{TOA}$ as compared to the observations. However, for CMIP6/s models, this underestimation of the OLR trend can be interpreted as a manifestation of simulated internal variability.

To motivate a deeper perspective on clouds, boundary layer dynamics, and advective heating at process level, we conduct a large sample of small-domain daily LES complementing the MOSAiC full drift. Averaged over all cases, radiative cooling of the full atmospheric column is balanced by advective heating (RAE). However, for liquid-bearing cloudy boundary layers the entrainment heating of the boundary layer is significant, even more important than horizontal advection, despite only modest efficiency in counteracting the radiative cooling. This links to Fig. 2c, which implies a negative contribution of clouds to the LRF in the lower ABL (extending up to 850 hPa), but a positive contribution in higher layers. We draw two conclusions from this result: Firstly, entrainment heating represents a significant column-internal redistribution of heat, which impacts the lapse-rate over sea ice. Secondly, liquid-bearing cloudy boundary layers are not in strict RAE, but are closer to RCE due to the significance of top-down convective heating through entrainment. However, a full equilibrium is not reached, indicating that these liquid-bearing low level air masses are still in the process of cooling. How these low level processes in effect contribute to AA requires further research, for example by conducting LES for perturbed climate conditions.

Beyond the processes analysed in the present study, also the rate at which sea ice melted in the study period plays a large role for simulated AA and LRF. However, we do not find a strong relation between skill in simulating sea ice and its decline, and the magnitude of AA within a model. CMIP6 models that are identified as capable of simulating a realistic amount of sea-ice loss together with a plausible change in global mean temperature over time (1979–2014; Notz and the SIMIP Community, 2020) span across our collection of models (acronyms marked in bolt in Table 1).

## 5 Conclusions

We have presented a variety of Arctic-based observations and reanalyses, in conjunction with projections of state-of-the art climate models within CMIP6 to find synergy among them in support of advancing our understanding of AA and the Arctic LRF. We propose a constraint on past AA and ALRF by attributing observable aspects of the current climate system to co-located CMIP6 simulations by models that project an either weak or strong AA and ALRF during past decades. In the scope of our main hypotheses formulated in point 1–3 of the introduction, we conclude the following key results, which largely focus on seasonal results during boreal fall and winter (most defining for both AA and ALRF):

1. Our data sets for boreal winter (and fall) show that the vertical temperature structure of the Arctic boundary layer is more realistically depicted in climate models with weak simulated AA/ALRF in the past. The attribution of observations to CMIP6/w models during DJFM is mainly based on data collected during the MOSAiC expedition (representing the central Arctic during that time), and dropsonde measurements in the Fram strait over sea ice. The CMIP6/w models in particular simulate a stronger present-day inversion through less depletion in the past, and generate a smaller low-level warming through sea ice retreat. The latter implies for weak-AA/ALRF models, less warming close to the surface for a given amount of sea-ice retreat, and thereby a smaller contribution to the positive ALRF through this process. More specifically, weak-AA/ALRF models remain weak-AA/ALRF models in future scenarios in this context.

2. The analysis of Pan-Arctic atmospheric transport convergence within the polar cap supports these constraints: This remote aspect that can further mediate the warming structure in the free troposphere is more realistically represented by climate models with weak simulated AA/ALRF in the past during fall and winter. In particular, models with weaker past AA/ALRF systematically simulate a smaller present-day atmospheric energy transport convergence in the Arctic during boreal fall and winter, which is consistent with reanalyses. We further explore changes in leading transport pathways that mediate vertically non-uniform warming, namely bottom and top-heavy warming. Albeit no clear attribution of either CMIP6/w or CMIP6/s to reanalysis results is possible, we highlight the potential of establishing links between large-scale regulated vertical warming structures, and the spatial distribution of Arctic feedbacks.

3. Lastly, we show the difference in temperature profiles and surface energy budget between cloudy and clear-sky conditions, in CMIP6 models, and LES data, respectively. Both climate models and LES simulations show that in cloudy cases, the vertical mixing becomes an important heating term for the atmospheric boundary layer. Even though we do not engage in a deeper study to attribute the representation of these processes to either weak or strong AA/ALRF simulations in the past, we want to motivate a perspective on the role of clouds on boundary layer dynamics and vertical warming structures. These processes are notoriously under-represented in literature concerning the Arctic LRF, but both local energy budget and vertical heat distribution can play an important role in its evolution.

*Code availability.* The LES code (DALES) used in this study is open access and available on GitHub at *https://github.com/dalesteam/dales*. The current version of DALES (*https://doi.org/10.5281/zenodo.5642477* : dales 4.3 with extension for mixed-phase microphysics) is available on github as *https://github.com/jchylik/dales/releases/tag/dales4.3sb3cgn*.

*Data availability.* The CMIP6 data are available from the Earth System Grid Federation (ESGF) system: https://esgf-node.ipsl.upmc.fr/projects/esgf-ipsl/, ESGF, 2022 (Eyring et al., 2016). GISTEMP data are available from https://data.giss.nasa.gov/gistemp/, Berkeley Earth data from http://berkeleyearth.org/data/, HadCRUT5 data from https://www.metoffice.gov.uk/hadobs/hadcrut5/data/current/download.html, NOAA data from https://www.ncei.noaa.gov/data/, and ERA5 data from https://cds.climate.copernicus.eu. Data from radiosondes launched during the MOSAiC expedition can be downloaded from PANGEA: https://doi.pangaea.de/10.1594/PANGAEA.928656 (Maturilli et al.,

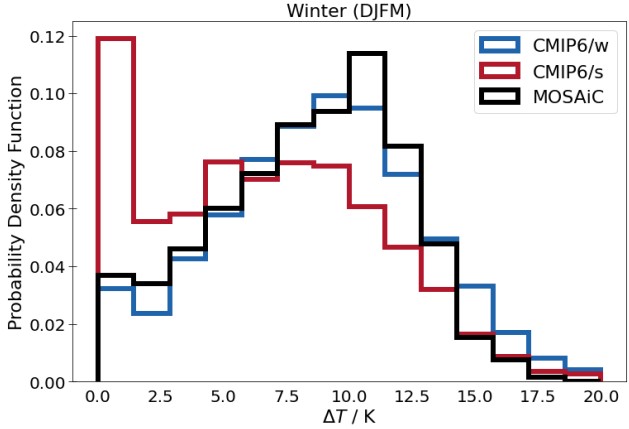

**Figure A1.** Supplement to Fig. 4: Histogram of inversion strengths $\Delta T$ obtained from MOSAiC-launched radiosonde, and for the model subsets CMIP6/w and CMIP6/s during DJFM.

2021). Near-surface temperature data during MOSAiC can be downloaded from the Arctic Data Center: https://arcticdata.io/catalog/view/ doi:10.18739/A2VM42Z5F (Cox et al., 2021) NSA radiosonde data are available at the DOE ARM data repository https://adc.arm.gov (Jensen et al., 1998). Dropsonde data from the three different aircraft campaigns considered in our study can be downloaded from the PANGAEA repository: Lüpkes and Schlünzen (1996); Lüpkes et al. (2021); Becker et al. (2020). ERA5 data can be downloaded from the ECMWF data catalogue: https://apps.ecmwf.int/data-catalogues/era5/?class=ea (Hersbach et al., 2020). AVHRR OLR data are available at

the DWD website: https://doi.org/10.5676/DWD/ESA_Cloud_cci/AVHRR-PM/V003. NOAA/NCEI HIRS OLR data can be downloaded at https://doi.org/10.24381/cds.85a8f66e.

## Appendix A: Temperature inversions from radio soundings during MOSAiC - DJFM

## Appendix B: Internal variability

*Author contributions.* The study was conceived by O.L. and J.Q. with contributions by all authors. O.L. performed the CMIP6 model
analyses and many of the model-to-data comparisons. C.S. contributed the feedback analysis tools and advised on their application and interpretation. B.V. and O.L. analysed the MOSAiC radiosondes vs. CMIP models with substantial help by M.D.S. and S.D. The Utqiaġvik analysis was from P.S.G., F.B. and H.K.-L. together with O.L.. S.B., A.E. and M.W. contributed the ropsonde analysis together with O.L.. The transport patterns were analysed by D.H. with input from O.L.. The energy transport is contributed by S.M. and C.J. with support by O.L.. R.N., J.C. and N.S. performed and analysed the LES; M.V., L.L. and K.V. analysed the satellite retrievals. G.S. advised on the role of
sea ice. All authors contributed to the manuscript writing.

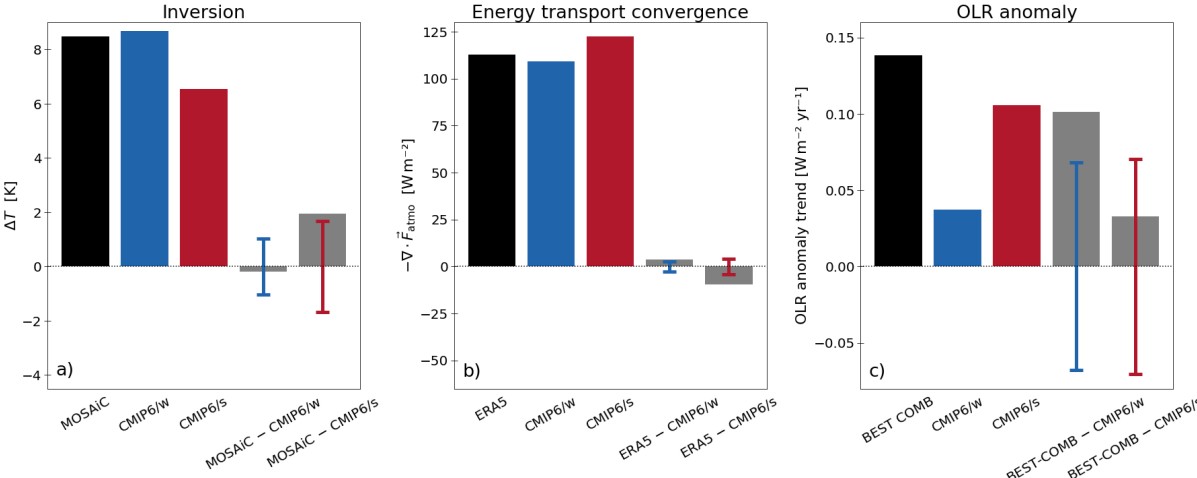

**Figure B1.** The role of internal variability: a) Averaged temperature inversion during DJFM for MOSAiC (black), and co-located CMIP6/w (blue) and CMIP/s (red) model data. Model data are expressed as ensemble means over all available realizations per subset. Gray bars give the residuals after subtracting the externally forced simulations (CMIP6/w and CMIP6/s ensemble means) from the observed inversion. The error bars indicate the 95 % range of simulated internal variability for both CMIP6/w (blue) and CMIP6/s (red) models, respectively. b) and c) are analogue to a), but for comparing observations/reanalyses of atmospheric energy transport convergence, and OLR anomaly trend to co-located CMIP6 data, respectively.

*Competing interests.*   The authors declare no competing interests.

*Acknowledgements.*   We gratefully acknowledge the funding by the Deutsche Forschungsgemeinschaft (DFG, German Research Foundation) – Projektnummer 268020496 – TRR 172, within the Transregional Collaborative Research Center "ArctiC Amplification: Climate Relevant Atmospheric and SurfaCe Processes, and Feedback Mechanisms (AC)[3]". M.D.S. was supported by a Mercator Fellowship as part of (AC)[3]. The authors would like to thank Benjamin Kirbus for constructive comments on the manuscript.

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

**Table 1.** All CMIP6 models used in this study with AA and ALRF derived from the surface-near atmospheric temperature, and lapse-rate difference, respectively, between 1985–2014 and 1951–1980. Table 1 further gives the time resolution available in the model diagnostics (6h: 6-hourly, day: daily, mon: monthly), together with the categorisation as weak or strong-AA models (CMIP6/w or CMIP6/s; in the superscript of the acronyms) per time-resolution group. Bold model acronyms indicate models that are most skilled at simulating a realistic amount of sea-ice loss together with a plausible global temperature change over time according to Notz and the SIMIP Community (2020).

| | Model acronym | AA / K | ALRF[*] / K | Time resolution | Reference |
|---|---|---|---|---|---|
| 1 | INM-CM5-0[day, CMIP6/w] | 0.210 | $0.078 \pm 0.015$ | day, mon | Volodin et al. (2019a) |
| 2 | INM-CM4-8[day, CMIP6/w] | 0.241 | $0.104 \pm 0.016$ | day, mon | Volodin et al. (2019b) |
| 3 | **GFDL-ESM4**[mon, CMIP6/w] | 0.274 | $0.120 \pm 0.022$ | mon | Krasting et al. (2018) |
| 4 | HadGEM3-GC31-LL[day, mon, CMIP6/w] | 0.549 | $0.167 \pm 0.032$ | day, mon | Ridley et al. (2018) |
| 5 | SAM0-UNICON[6h, mon, CMIP6/w] | 0.552 | $0.313 \pm 0.053$ | 6h, mon | Park and Shin (2019) |
| 6 | **MPI-ESM-1-2-HAM**[6h, CMIP6/w] | 0.641 | $0.208 \pm 0.041$ | 6h, day, mon | Neubauer et al. (2019) |
| 7 | CMCC-CM2-HR4 | 0.642 | $0.305 \pm 0.051$ | mon | Scoccimarro et al. (2020) |
| 8 | **ACCESS-CM2** | 0.668 | $0.203 \pm 0.038$ | mon | Savita et al. (2019) |
| 9 | MIROC-ES2L | 0.678 | $0.241 \pm 0.037$ | mon | Hajima et al. (2019) |
| 10 | AWI-ESM-1-1-LR[6h, CMIP6/w] | 0.689 | $0.309 \pm 0.045$ | 6h, day, mon | Danek et al. (2020) |
| 11 | **NorESM2-MM** | 0.695 | $0.264 \pm 0.041$ | 6h, mon | Bentsen et al. (2019) |
| 12 | CESM2-FV2 | 0.729 | $0.197 \pm 0.039$ | day, mon | Danabasoglu (2019a) |
| 13 | **BCC-CSM2-MR** | 0.743 | $0.281 \pm 0.037$ | mon | Xin et al. (2018) |
| 14 | CNRM-CM6-1 | 0.743 | $0.326 \pm 0.043$ | 6h, day, mon | Voldoire (2018) |
| 15 | **MPI-ESM1-2-LR** | 0.751 | $0.282 \pm 0.042$ | 6h, day, mon | Wieners et al. (2019) |
| 16 | MIROC6 | 0.769 | $0.277 \pm 0.044$ | 6h, mon | Tatebe and Watanabe (2018) |
| 17 | ACCESS-ESM1-5 | 0.787 | $0.234 \pm 0.043$ | mon | Ziehn et al. (2019) |
| 18 | **GISS-E2-1-G** | 0.814 | $0.343 \pm 0.050$ | 6h, mon | NASA/GISS (2018a) |
| 19 | UKESM1-0-LL | 0.817 | $0.294 \pm 0.042$ | day, mon | Tang et al. (2019) |
| 20 | NESM3 | 0.824 | $0.330 \pm 0.050$ | mon | Cao and Wang (2019) |
| 21 | **MPI-ESM1-2-HR** | 0.830 | $0.343 \pm 0.048$ | 6h, day, mon | Jungclaus et al. (2019) |
| 22 | CESM2-WACCM-FV2 | 0.867 | $0.271 \pm 0.045$ | day, mon | Danabasoglu (2019b) |
| 23 | GFDL-CM4 | 0.875 | $0.293 \pm 0.048$ | day, mon | Guo et al. (2018) |
| 24 | CESM2-WACCM | 0.933 | $0.296 \pm 0.054$ | day, mon | Danabasoglu (2019c) |
| 25 | CNRM-ESM2-1[6h, CMIP6/s] | 0.956 | $0.410 \pm 0.055$ | 6h, day, mon | Seferian (2018) |
| 26 | **FGOALS-f3-L** | 0.960 | $0.464 \pm 0.066$ | mon | Yu (2018) |
| 27 | CESM2 | 0.993 | $0.334 \pm 0.058$ | day, mon | Danabasoglu (2019d) |
| 28 | **CNRM-CM6-1-HR**[6h, day, CMIP6/s] | 1.002 | $0.392 \pm 0.041$ | 6h, day, mon | Voldoire (2019) |
| 29 | IPSL-CM6A-LR[6h, day, mon, CMIP6/s] | 1.062 | $0.430 \pm 0.070$ | 6h, day, mon | Boucher et al. (2018) |
| 30 | **MRI-ESM2-0**[day, mon, CMIP6/s] | 1.116 | $0.380 \pm 0.060$ | day, mon | Yukimoto et al. (2019) |
| 31 | GISS-E2-1-H[mon, CMIP6/s] | 1.148 | $0.533 \pm 0.064$ | mon | NASA/GISS (2018b) |

[*] ALRF values are computed by averaging the results derived from several kernels (CAM5, GFDL AM2, ERA-Interim, HadGEM3-GA7). The inter-kernel standard deviation gives the error range.