# Peer review of "Constraints on simulated past Arctic amplification and lapse-rate feedback from observations"

_Atmospheric Chemistry and Physics, 2022_

## Author Comment (AC1)

**Author's response to RC1**

The authors combine CMIP6 model output with reanalysis data, observations and LES model results to investigate the inter-model spread in Arctic amplification (AA) and the Arctic lapse-rate feedback (ALRF). When sorting models into models with stronger and weaker AA and ALRF, strong AA/LRF models better match reanalysis trends in heat advection, whereas weak AA/ALRF better match observed present-day inversion strength. The presented data and work is interesting and relevant to important research questions, but I have a few major concerns on how the model-observation analysis is carried out.

**Reply**: We thank Reviewer 1 for very constructive and helpful comments on our manuscript. We have addressed the concerns which have helped us to improve our manuscript. Our responses are listed below.

**Major**

**Reviewer Point P 0.1** — The authors do not investigate the role of internal variability for model results. Investigating only one ensemble member per model without regard for the ensemble spread might not do justice to models – even a clear mismatch with observations does not rule out that the model in question is consistent with the observed trend or phenomenon (see e.g. Notz, 2015)

**Reply**: The reviewer raises an important point. Firstly, we haven't been clear enough in elaborating on the use of different ensemble members within CMIP6. We use the entire data set (all available ensemble members), but as ensemble means over all realisations per model – this way, each model carries equal weight in the CMIP6 distribution, and we exclude the chance of accidentally choosing a model realisation that deviates substantially from the entire population The reviewer still is right, we do not account for internal variability by using ensemble means, we merely exclude the chance of catching an outlier among the realisations. We want to state that while internal variability is an important and very interesting point, unfortunately there are only few models with enough members to engage in a deeper study: Only four models have more than 30 realisations which could be considered enough for such an analysis. Six other models on the other hand only have only one realisation, more than half only 2–3. Since we noticed that this topic has not been addressed properly in our manuscript, we added a paragraph in the methods section (L134 ff) and thank the reviewer for pointing it out.

**Reviewer Point P 0.2** — Important conclusions rely on small subsets of the analysed models, comparing only the top and bottom three models in terms of AA/ALRF. For the weak AA group, these are clear outliers in the CMIP ensemble, and two of the three are different versions of the same model. Would the results remain the same (just with weaker signals) if models 4-8/24-28 were used instead?

**Reply**: It is an important question that the reviewer asks here. Firstly, it is exactly that, we chose CMIP6 models at the respective edges of the range of simulated past AA. This is to ensure a clear signal in the comparison and to allow for an attribution to either weak and strong-AA models (we added a comment in the method section L146 and following). Since we don't take the classic approach of an emergent constraint where statistically strong relationships across model simulations of past/future

and the observable current climate are used, we instead agreed on a number of models that represent an either weak or strong-AA cluster (split by the observed value of AA, also added now in Fig. 1 and explained in the text L151 and L420 and following). Unfortunately, some comparisons required a high temporal resolution of the model output (L140 ff in the manuscript). The model-data comparison at 6-hourly time resolution in particular included only 12 models with all required diagnostics in total (Section 3.2-3.4 concerning stability and vertical temperature structures). This has lead us to the compromise of choosing 3 models at the respective edge of AA distribution (model 5, 6, 10 for weak, and 25, 28, 29 for strong), as the inclusion of more models would rather represent the inter-model mean.

However, we strongly agree with the point being raised. To demonstrate that the comparison is still valid, we added model 11 to the weak-AA, and model 21 to the strong-AA ensemble to extend the individual range. This has no effect on the key messages of Section 3.2-3.4, the results remain the same (e.g. shown for MOSAiC in Fig. R 1). We added a paragraph on the sensitivity of model choice on the results in the discussion part L734 ff. In addition, the model-to-data comparison is explained more thoroughly in Section 3.2, adding a supporting Figure to the Appendix, and further commenting on statistical representatives of the results that rely on sparse high-resolution model diagnostics.

[Figure]

Figure R 1: As Fig. 4 b in the manuscript, with the addition of model 11 and 21 to CMIP6/w and CMIP6/s subsets, respectively. In each panel, the left box plots show the original subsets with three models, and the right box plots show the subsets with 4 models, respectively.

**Reviewer Point P 0.3** — The definition of AA as a difference $dT_{Arctic}$-$dT_{global}$ rather than a ratio $dT_{Arctic}/dT_{global}$ is surprising to me. Wouldn't one expect most mechanisms driving AA to act in a multiplicative rather than additive way? Similarly, the choice of the reference period is unclear to me. If no observations from the reference period are used, why not choose an earlier reference period (PI or at least 1850-1880 historical) to maximize the signal?

**Reply**: We thank the reviewer for bringing up these points, which we have discussed in the preparation of the study also internally. To address the first point of defining AA: There are different metrics which can be used to describe the difference in temperature change between the northern high latitudes and the global (or mid-latitude / tropical) mean to quantify AA. There are several studies that apply different metrics, e. g., the difference between present and base climate (like us; e.g., Francis and Vavrus, 2015, the ratio, or ratio between linear trends (Johannessen et al., 2016; Kobashi et al., 2013). Indeed the ratio is an established metric, as the reviewer suggests, but there is no fundamental information it carries that the difference would not carry. The reason for choosing the difference is a practical one. When using the ratio of anomalies (e.g., here for the temperature) the denominator may approach small numbers down to zero. In the period of interest, for some model realisations, it turned out that global

warming is rather close to 0 (e.g. model 13 realisation r20i1p1f2 with 0.11 K global warming), so the ratio estimator may be arbitrarily inflating the model spread. A consequence in our study is that when using the ratio metric, the correlation between ALRF and AA degrades to $r = 0.66$, instead of $0.86$ as in Figure 1. We consider the LRF a stable metric to quantify AA as it has essentially the same physical basis: The feedback contributes to slight cooling on global average in the time period of interest, but strong warming in the Arctic, both of which is a result of the effect of strong vs. limited mixing abilities in the tropics vs. Arctic on the vertical redistribution of the warming. In the Arctic, this imposes the key feature of bottom-heavy warming, which is AA. Thereby, we chose the difference definition: first, it reduces the problem with small global-warming in some model runs, and second, we can make use of the stronger ALRF-AA relationship by classifying strong/weak AA models also as strong/weak ALRF models, by extension. We now added this explanation in Section 2.1. (L185 ff).

To address the second point of time framing: In our first analyses we did consider the entire time series of historical simulations. However, there are two main periods which are identified to have AA, and both occur in the 20th century: in the 1920–1940s, and at the end of the 20th century continuing into the 21st century (Davy et al., 2018 and references therein). We added this important information at the beginning of the introduction, and in the methods L189 ff. In addition, in Section 3.5, we actually address changes in the reference period (relative frequency of circulation regimes) in ERA5 data. This type of comparison is only feasible with reanalysis data, which starts from 1950. This has led us to adapt the reference time period. Another important point is that large changes in the global surface temperature (simulated and observed) have started to occur since the second half of the 20th century. This leads to the result that excluding the first century of historical simulations imposes no large impact on the order of models 1-31 which are sorted by the degree of AA. Short message: It does not matter for the outcome of this study if 1850–2014 or 1950–2014 is used, but it allows for the inclusion of Section 3.5, and it addresses the second (and stronger) period of identified AA.

**Minor**

**Reviewer Point P 0.4** — l 22 ff and elsewhere in the manuscript: Now that the work is done, I feel that the manuscript would be stronger by focusing on what has been achieved rather than what the authors want to achieve.

**Reply**: The last paragraph was omitted and L11-12 added "...to provide different perspectives on AA and the Arctic LRF." instead.

**Reviewer Point P 0.5** — l. 65 ff: The impact of clouds on the vertical temperature profile has not been introduced at this point in the manuscript.

**Reply**: That is true. We added a comment in L68 ff.

**Reviewer Point P 0.6** — l 205: showing that 2019/2020 is equivalent to 2000-2014 using scenario output would be stronger than just assuming it – strong changes have happened in the Arctic in the early 21st century.

**Reply**: We thank the reviewer for pointing to our insufficient elaboration on time comparison here. The comment is most valid for comparing the very recent and highly valuable MOSAiC data during 2019/2020 with the last years of historical simulations (2000–2014). Our choice of model data here

is, again, somewhat a result of data availability, which is unfortunately limited in the 6-hourly time resolution: Only three models from Table 1 of the manuscript provide the 6-hourly time resolution also in the scenario simulations ongoing from 2014. However, we acknowledge that this alone cannot justify a comparison here. We argue that our comparison is valid, and show a comparing time series between scenario data (SSP585 as upper boundary of the range of scenarios) for 2019–2020 (MOSAiC time frame) and historical data 2000–2014 for those models that provide both. Fig. R 2 shows that the SSP585 time series lies within the inter-annual range of the 2000–2014 period, and for most of the year, within the range of inter-annual standard deviation. Even though we cannot show this comparison for each model used in our study, we argue that the correspondence between 2000–2014 and 2019–2020 time series from the highest emission scenario justifies our comparison in Section 3.2. We added a comment in L233 ff.

[Figure]

Figure R 2: Comparing time series for surface-based temperature inversion $\Delta T$ for MOSAiC conduction time (2019–2020; SSP585 scenario in CMIP6), and for historical data 2000–2014, which we compare to the MOSAiC radiosonde data in Section 3.2 of the manuscript. Those models that facilitate the comparison are CNRM-CM6-1, MIROC6, and MRI-ESM2-0.

**Reviewer Point P 0.7** — For the comparison with radiosondes, I would recommend coarsening the radiosonde profiles to the vertical resolution of the models at least as a sensitivity test (same for NSA).

**Reply**: Unfortunately, the suggested sensitivity test is complicated here, since the model diagnostics are given on model levels. This would require interpolating the models profiles to common pressure levels in order to coarsen the radiosonde profiles to a common vertical resolution. Our approach is to keep each model on its instantaneous vertical resolution and derive the inversion as described in 2.2 and 2.3 of the manuscript. We now specify this approach in L227.

**Reviewer Point P 0.8** — Section 2.4: Comparing March/April measurements with DJFM model data – did you check that model data looks similar for March as for the entire winter season?

**Reply**: There might have been a misunderstanding due to an imprecise formulation on our side. We compare the flight campaign data exclusively during March with model data during the same month (not DJFM). We re-formulated the sentence "The measurements presented here were performed during March to ensure similar thermodynamic conditions compared to the extended winter season, DJFM." to: "Since the measurements presented here are available only for March, we restrict the model-observation-comparison to this month." (L269 ff).

**Reviewer Point P 0.9** — Do we expect the 1993 campaign to show the same climate state as the 2019 campaign?

**Reply**: The reviewer is right, between 1993 and 2019, the climate state is different. However, the comparing time period is 2000–2014, and the year-average of the aircraft campaigns lies within the range of model data: $\text{avg}(1993, 2013, 2019) \approx 2008$. We were still interested in the comparison without the 1993 campaign, but the results are similar. Only over ice, the inversion is slightly elevated and weaker (by around 1 K), which does not affect the conclusions, however. The warming effect by transforming from sea ice to ocean is less, compared to the combination of all campaigns, which brings the observations even closer to the CMIP6/w model ensemble. However, we prefer keeping as many data as possible for the observational constraint: When including too little data, it becomes more illusive to which extend our results are mediated by climate change or ambient meteorology. We thereby included the REFLEX data to achieve a wider range of conditions.

**Reviewer Point P 0.10** — l 385: do all models have similar inversion strengths in the reference period?

**Reply**: We thank the reviewer for bringing more attention to this comparison. The models within each subset do not have exactly the same strength, but both model groups show no overlap (weak-AA models 7.55–10.62 K, and strong-AA models 5.75–6.91 K during DJFM on average). Thereby, the subsets are clearly distinguishable, and the MOSAiC inversion average of 8.49 K lies in the range of CMIP6/w models. We added this important comment in L459 ff, and further elaborate on the statistical representatives of the comparison, primarily during the season of highest interest which is DJFM.

**Reviewer Point P 0.11** — l 407: what is the time frame covered by the Kahl (1990) study? Do we expect it to be representative of 2020 conditions?

**Reply**: Agreed, the mentioned study should not be used in this argumentation here, especially since we expect the inversion strength to decrease with time, which would explain stronger inversions observed in the study of 1990. We drop the reference and adapted the text accordingly.

**Reviewer Point P 0.12** — l 487: what significance level? How did you do the bootstrap analysis?

**Reply**: We now explain the bootstrap analysis more clearly in Section 2.5

**Reviewer Point P 0.13** — Fig. 10 and related analysis: This shows data year-round, is there a relevant seasonal cycle?

**Reply**: There are mild seasonal variations, however the two-state feature is evident throughout the year. A cloudless atmosphere is thereby in approximate RAE, and cloudiness adds a heat source to the boundary layer. This features confirms the results of Figure 2 of the manuscript, and is further in line with previous findings (e.g., Pithan et al., 2014). An explicit evaluation of the seasonality was not pursued here, as this plot is mostly an outlook and frame to the introducing Figure 2 (GCM results also confirmed by LES simulations).

**Reviewer Point P 0.14** — l. 564: Cronin and Jansen (2016) would be a good reference here.

**Reply**: Added here, and also in L602.

**Reviewer Point P 0.15** — l. 585–590: I think this is an important result deserving a stronger emphasis in the paper, since entrainment has not received a lot of attention in this context so far.

**Reply**: The reviewer is right that this is a very interesting result. However, the results are meant to give a final view and supplement to the introducing Figure 2 of the manuscript, rather than following the model-to-OBS/reanalysis framework as the other sections. Therefore, we do not want to overemphasise the point here. However, the implication is clear: entrainment warming due to the presence of clouds is a considerably large heat source for the surface, and the presence of clouds might therefore reduce the change in lapse rate in the lower boundary layer (as already suggested from climate models in Fig. 2). This is an important result for understanding the LRF, and leaves room for deeper studies, not only due to the under-representation of the role of clouds when studying the LRF. We motivate the importance of clear vs. cloudy states in the discussion, but do not further dig into the results here, since this point deserves a dedicated study on its own and would inflate our study at the moment, (rather, dedicated studies are underway).

**Reviewer Point P 0.16** — l. 592 "we compile a sizeable amount of observations" Here and elsewhere in the paper: There is nothing to be said against impressing the reader with the large array of observations you bring to the task in addition to CMIP and LES data, but in my view this works better if you leave being impressed to the reader.

**Reply**: Changed in "We have presented data from several Arctic-based observations and reanalyses in conjunction with co-located CMIP6 model simulations to constrain various processes that mediate both Arctic amplification and the Arctic LRF." Also, the abstract is adapted according to the suggestion.

**Reviewer Point P 0.17** — l. 687: I think a crucial point here is that CMIP6/s models generate less warming for a given amount of sea-ice retreat. If this is correct, it should be stated more explicitly.

**Reply**: It is actually the opposite: Weak-AA models have a stronger present-day inversion (over both sea ice and open ocean), and when transforming from sea ice to open ocean, the expected warming of the lower boundary layer is less compared to CMIP6/s. We now state this more clearly in point 1 of the conclusions.

**References**

Davy, R., Chen, L., and Hanna, E. (2018). Arctic amplification metrics. *International Journal of Climatology*, 38(12):4384–4394.

Francis, J. A. and Vavrus, S. J. (2015). Evidence for a wavier jet stream in response to rapid arctic warming. *Environmental Research Letters*, 10(1):014005.

Johannessen, O. M., Kuzmina, S. I., Bobylev, L. P., and Miles, M. W. (2016). Surface air temperature variability and trends in the arctic: new amplification assessment and regionalisation. *Tellus A: Dynamic Meteorology and Oceanography*, 68(1):28234.

Kobashi, T., Shindell, D., Kodera, K., Box, J., Nakaegawa, T., and Kawamura, K. (2013). On the origin of multi-decadal to centennial greenland temperature anomalies over the past 800 yr. *Climate of the Past*, 9(2):583–596.

Notz, D. (2015). How well must climate models agree with observations? *Philosophical Transactions of the Royal Society A: Mathematical, Physical and Engineering Sciences*, 373(2052):20140164.

Pithan, F., Medeiros, B., and Mauritsen, T. (2014). Mixed-phase clouds cause climate model biases in arctic wintertime temperature inversions. *Climate dynamics*, 43:289–303.

---

## Author Comment (AC2)

**Author's response to RC2**

We thank Reviewer 2 for very constructive and helpful comments on our manuscript. We have addressed the concerns which have helped us to improve our manuscript. Our responses are listed below.

**Major**

**Reviewer Point P 0.1** — This paper aims to dissect the Arctic warming simulated in the CMIP6 models by comparing them to observations. The analysis is centered on the geophysical variables related to the lapse rate feedback, which, as argued by a number of studies, is of critical importance for the Arctic warming amplification. To the extent this argument is valid, the comparisons in this paper are well motivated. A novel aspect of this paper is that it includes comparisons to several different kinds of data, some of which, such as the newly acquired Mosaic campaign data, provides fresh perspectives for model validation. However, although each comparison included here potentially provides a useful line of evidence for discriminating the models, unfortunately few results appear conclusive in the end. This calls into question whether one had better aim to identify and focus on what can be more conclusively stated about the models and/or nature, as opposed to a somewhat nonselective listing of results.

**Reply**: We acknowledge the major criticism that is being raised here, and we would have also hoped for a more clear story in parts. However, we want to justify presenting each of the results in this study: Firstly, to some extent and especially for the model comparison to the observations at higher time resolution, we do not have clear emerging relationships between simulated AA/ALRF and present-day climate aspects that is used to constrain the mediating processes. This is simply due to the fact that only few simulations exist, even in the historical simulations, to derive a relationship as done for e.g., an emergent constraint. However, especially these local processes are crucial in better understanding and constraining both ALRF, and AA as a whole. Our idea was to chose those models at the edge of the AA/ALRF distribution as a compromise. If the observational constraint then fits either one of these categories, we have a clear signal. If the OBS are in the range of inter-model mean, then the attribution to either weak or strong-AA/ALRF is less straightforward. This might appear as somewhat inconclusive, but it is still a result, i.e., an attribution to the inter-model mean. That is why we want to show all results, also to cover each process that is believed to have a mediating impact (inversion, sea ice retreat, transport, ...) and reduce gaps in the interpretation. We prefer not to select some that match one line of evidence and omit others. The collaborative project continues and in the next project phase one aim is to reconcile the differences in conclusions.

We adapted several major changes to the manuscript to clarify our intention: At the end of each section, we now comment more carefully on the significance in model differences, and the attribution of several observations to either one of the emerging subsets, just as the synergy that gradually appears trough the result sections. These inter-mediate results are later brought into context: The conclusion focuses only on results that show a clear signal in both model subset differences and their constraint through observations, and further brings attention to the synergy that emerges between inter-mediate results.

**Reviewer Point P 0.2** — Moreover, the use of some data and analysis methods are not sufficiently explained (see comments below), raising questions about their properness. For these reasons, I think

the paper would need a major revision before being considered for publication.

**Reply**: We thank the reviewer for their specific suggestions and performed the major revisions.

**Reviewer Point P 0.3** — Figure 1. Can you also provide the observations for a comparison in these diagnostics?

**Reply**: It is a very good suggestion to add the observed estimate for AA 1985–2014 with respect to 1951–1980. We added the average from several observational estimates to Figure 1 of the manuscript. We present the OBS estimate of AA in Figure 1 of the manuscript as average from several observational data set (GISTEMP, Berkeley Earth, HadCRUT5, NOAA's MLOST, and ERA5; Rantanen et al., 2022). An overview of the observed AA as time series is shown in Fig. R 1. We added comments on the OBS estimate in Section 2 L151 ff. In addition, the inclusion of OBS in the introducing plot allows us to interpret the simulated model range with respect to observations. It ensures that our classification of either weak or strong-AA model subset actually shows AA values below or above the OBS, respectively. We include this interpretation in Section 3.1. L420 ff, and additionally expand the elaboration on statistical significance in the discussion. This concerns primarily the previously mentioned model-to-OBS comparisons at 6-hourly time resolution (MOSAiC, NSA, dropsondes), which is limited by the availability of models. Simply categorising the model range by taking the top-3 lowest and highest AA models might not do justice to the classification as either weak or low AA simulations at 6-hourly resolution, since the entire model spread (all models in Table 1 of the manuscript) is larger. However, by adding the observations we show that the discrimination is still valid, since the sub-set average of AA for CMIP6/w, and CMIP6/s lies below, and above the OBS estimate, respectively (for any time-resolution group). This gives further justification to our approach.

[Figure]

Figure R 1: Time series of AA: Difference in annual mean temperature anomalies in the Arctic with respect to global average as derived from the various observational datasets. Temperature anomalies have been calculated relative to the 30-year period of 1951–1980.

**Reviewer Point P 0.4** — L172 "consistency": can you provide any reference to this belief? Note that it is quite known that there are noticeable differences between different kernels, especially in the Arctic. In either case, it would be move convincing to provide an error bar based on results computed from more than one kernel.

**Reply**: We agree with the reviewer that the formulation "consistency" is too unspecific in this context. We show in Fig R 2 the same scatter plot as in Figure 1 b of the manuscript, but with ALRF values derived from different kernels (the inter-model distribution of AA is not effected by the choice of kernel). There are indeed differences in the quantification of the ALRF across the kernels, and slight variations of the inter-model correlation between AA and ALRF. We acknowledge the criticism being raised from the reviewer and now show the scatter plot in Fig. 1 b, but with model-specific ALRF values derived as average from the output of all kernels. To avoid making Fig. 1 even more busy, we account for the inter-kernel spread across ALRF by adding the standard deviation in Table 1. We further added a comment on method Section 2.1 L192 ff. Albeit there are difference in the relationship of the inter-model spread in AA and ALRF across CMIP6 models, we emphasise that our results are not sensitive to the choice of kernels. The classification as either weak or strong-AA models remains unaffected, and the AA-ALRF relationship increases even for other kernels than the previously chosen HadGEM3 kernel. Thereby, the attribution of weak/strong-AA models to equally weak/strong-ALRF models is still valid. The newly added comment proves that point, and is important for the credibility of our results.

[Figure]

Figure R 2: As Fig. 1 b of the manuscript, but with different kernels to derive the ALRF.

**Reviewer Point P 0.5** — L205, 251 use of years of 2010–2014. Can you justify the use of these model years to match the observation? It's understood coupled model years are nominal but what guarantees a comparison done here, between a single realization of nature of limited length and multiple model years, is proper? Very handwavy to "assume" they're "roughly the same".

**Reply**: We thank the reviewer for pointing to our insufficient elaboration on time comparison here. This comment is most valid for the limited data comparison at 6-hourly time resolution and concerns mostly the evaluation of model data based on very recent MOSAiC data (2019–2020). We have been considered using data from CMIP6 scenarios to expand the simulation period to the years following the historical simulations. However, this was again limited by the availability of data: Only three models in Table 1 of the manuscript provide the required diagnostics for simulation scenarios ongoing from 2014, which

was not an option. We were still highly interested in comparing climate models against the valuable data conducted during the MOSAiC expedition. It is true that the time shift between 2000–2014 and 2019–2020 raises questions about the validity of the comparison. To prove that it is still valid to treat this periods as part of the same climate state, we show for the three models with scenario output the time series comparison between 2000–2014 and 2019–2020 in Fig. R 3. We use scenario outputs from the highest emission scenario SSP585 as boundary of the range of scenarios for 2019–2020. Even for this highest scenario, the 2019–2020 time series lies within the inter-annual range of the 2000–2014 period, and for most of the year, within the range of inter-annual standard deviation. Even though we cannot show this comparison for each model used in our study, we argue that the correspondence between 2000–2014 and 2019–2020 time series from the highest emission scenario justifies our comparison in Section 2.2 We added a comment in L233 ff.

[Figure]

Figure R 3: Comparing time series for surface-based temperature inversion $\delta T$ for for MOSAiC conduction time (2019–2020; SSP585 scenario in CMIP6), and for historical data 2000–2014, which we compare to the MOSAiC radiosonde data in Section 3.2 of the manuscript. Those models that facilitate the comparison are CNRM-CM6-1, MIROC6, and MRI-ESM2-0.

**Reviewer Point P 0.6** — L261 The identification of different "regimes" looks an interesting approach to me. However, I found the description of the method too brief here. I'd suggest showing the relevant results such as the EOFs, as well as the associated PCs and eigenvalues. I think this method, like the other data and methods in this paper, is worth more careful/critical reasoning and more thorough discussion.

**Reply**: For this part of the study, we have used the concept of atmospheric circulation regimes (e.g., Hannachi et al., 2017) to characterise the large-scale circulation in terms of a few preferred states. This concept provides a framework for understanding low-frequency variability due to transitions between different regimes. In addition, Palmer (1993, 1999) introduced a dynamical paradigm for climate change which suggests, that a weak external forcing does not change the structure and number of atmospheric regimes, but instead changes the frequency of occurrence of the regimes. Since then, many studies have analysed the atmospheric circulation within this concept (see extended review by Hannach et al., 2017). To follow the reviewer's advise, we extended the description of the method for the determination of the regimes in section 2.5: L300 ff.

To characterize the reduced state space, we show here the spatial structure of the five leading EOFs over the North-Atlantic-Eurasian region (Fig. R 4a, 57.5% explained variance) and over the North-Pacific region (Fig. R 4 b, 54.5% explained variance) based on ERA5 daily mean SLP anomaly fields for the extended winter season (DJFM). The leading EOFs resemble the well-known teleconnection

[Figure]

Figure R 4: **a)** Left from top to bottom: Five leading EOFs over the North-Atlantic-Eurasian region for DJFM, based on ERA5 daily mean SLP anomalies for DJFM, explaining 17.5 %,14.3 %,11.5 %, 8.9 %, 5.4 %. **b)** Left from top to bottom: Five leading EOFs over the North-Pacific region for DJFM, based on ERA5 daily mean SLP anomalies for DJFM, explaining 16.3 %, 13.7 %, 11.0 %, 7.2 %, 6.6 % of the total variance respectively. Right from top to bottom: Corresponding time-series of principal components (normalized), the red line represents running mean values over 10 years.

patterns such as the North-Atlantic Oscillation (North-Atlantic EOF1), Scandinavia pattern (North-Atlantic EOF2), East Atlantic pattern (North-Atlantic EOF3), Pacific/North American pattern (North-Pacific EOF1), West Pacific pattern (North-Pacific EOF2).

**Reviewer Point P 0.7** — L290 What's the basis of using this proxy as a quantitative measure of the energy transport? How can the TOA-only perspective differentiate atmospheric vs. oceanic transports? How is equilibrium verified, so that horizontal transport can be inferred from vertical energy flux?

**Reply**: We thank the reviewer for commenting on the derivation of the transport term and acknowledge that the method has not been explained sufficiently at this point. In addition, we understand that it is useful to exclude the ocean signal while looking at atmospheric processes. Therefore, we now present the transport term as contribution from the atmosphere, and as divergence term wihtin an energy budget framework: Following previous works of e.g., (Nakamura and Oort, 1988; Trenberth, 1997; Serreze et al., 2007) we can consider the energy budget of an atmospheric column that extends from the surface to

the TOA. For each column, the tendency in energy storage within an atmospheric column $E_\mathrm{a}$ can be estimated as

$$\frac{\partial E_\mathrm{a}}{\partial t} = R_\mathrm{a} + Q_\mathrm{H} - \nabla \cdot \vec{F}_\mathrm{a}, \tag{1}$$

with net atmospheric radiation budget $R_\mathrm{a}$, the sum of turbulent heat fluxes at the surface $Q_\mathrm{H}$, and the convergence of the horizontal atmospheric energy transport $-\nabla \cdot \vec{F}_\mathrm{a}$. In the long-term and large-scale energy budget, we can further neglect the storage tendency under assumption of steady state (Serreze et al., 2007; Linke and Quaas, 2022). We use this simplified energy budget framework to estimate the horizontal convergence of energy transport indirectly, i.e., residual of the budget equation. According to the reviewers comment, we expanded the method description in Section 2.6, specifically clarifying that we exclusively consider the atmospheric transport (convergence), and further commenting on the equilibrium criteria. Both results and discussions are updated accordingly.

**Reviewer Point P 0.8** — L305 and Figure 9, concerning the use of satellite OLR records, it should be noted that various issues had been documented on how wrong it could be to take a non-SI-traceable radiation record as the ground-truth of "observed" long-term trends. For example:
Trishchenko et al. https://doi.org/10.1029/2002JD002353
Wong et al. https://doi.org/10.1175/JCLI3838.1
The OLR trending itself would be worth a full section if not a paper by itself. Before its correctness is established, it is very questionable to use this result as a model discrimination metric.

**Reply**: It is an important point being raised how to define the ground-truth of our data set. We want to comment on the credibility of the AVHRR climate data record: It is well known that suboptimal radiometric calibration of the AVHRR thermal channels might lead to inconsistencies, then source of discrepancies in Arctic cloud detection and radiation fluxes at the surface (Zygmuntowska et al., 2012). Prior to the production of the satellite record, every PM sensor was cross-calibrated with well-behaved sensors. The SCanning Imaging Absorption spectroMeter for Atmospheric CHartographY (SCIAMACHY) served as spectral reference for the visible wavelengths and the Infrared Atmospheric Sounding Interferometer (IASI) for the thermal channels (Stengel et al., 2020). This resulted in an improvement of the retrieved cloud parameters (Sus et al., 2018; McGarragh et al., 2018) in terms of precision, accuracy and stability (Stengel et al., 2017). Specifically to the Arctic, the AVHRR cloud record has not shown any scale-dependent bias upon validation with coincident measurements at four high-latitude ground sites (Vinjamuri et al., 2023). The accuracy of the cloud record is a key factor because the cloud properties are input for the derivation of the broadband fluxes (Henderson et al., 2013). The resulting accuracy in AVHRR-derived OLR amounts to $\pm 3\,\mathrm{W\,m^{-2}}$ against observations of the Geostationary Earth Radiation Budget (GERB) radiometer on board the Meteosat Second Generation (MSG-2) satellite (Christensen et al., 2016). This value is within GERB's calibration limits for radiation at TOA (Clerbaux et al., 2009). In relative terms, the average long-term bias of AVHRR-derived outgoing LW fluxes against CERES amounts to $-2.7\%$ (Stengel et al., 2020). In addition, using the same algorithm for the broadband fluxes, but applied at CloudSat, CALIPSO, and MODIS measurements instead, Kay and L'Ecuyer (2013) quantify an average bias against the Clouds and the Earth's Radiant Energy System (CERES) Energy Balanced and Filled (EBAF) record (Kato et al., 2018; Loeb et al., 2018) of the order of 4-5 $\mathrm{W\,m^{-2}}$. Consequently, the present AVHRR record has been used for the analysis of Arctic cloud radiative forcing (Lelli et al., 2023) and feedback (Philipp et al., 2020).

However, we acknowledge the comment of the reviewer and add additional data sources as reference to compare against CMIP6 models. The additional data records are the OLR flux from NOAA/NCEI

from the High Resolution Infrared Radiation Sounder (HIRS) instruments on board the NOAA and MetOp satellites, and ERA5 reanalyses. Additionally, we adapt the data record in Figure 9 of the manuscript to display the anomaly of the OLR with respect to the first 15 years of the AVHRR record (1983–1997). We apply this change since the main focus is the trend in OLR during recent decades, which is more straightforward in the new plot version. We further consider CERES satellite data in the absolute time series (not shown), which fits well the records derived for HIRS, AVHRR, and ERA5. Only in the anomaly plot as now presented in the manuscript, CERES does not appear due to insufficient time coverage (start 2000).

**Reviewer Point P 0.9** — L380 "significant". Although significant differences are stated here and at multiple other places (in this (Figure 4) and other figures), looking through these results, I am not convinced there is indeed any strong difference between the compared groups, either between "w" vs "s" or between them and the observation (Mosaic). If the discriminations are based on such weak evidence, I am not sure the observation used here provides any useful constraint as wished by the authors, or any model evaluation result can be considered conclusive. Please critically review and reason about this and other conclusions.

**Reply**: We thank the reviewer for pointing that out. In the specific case, and also in the two following analyses including 6-hourly data, we partly omitted the word "significance" since it implies the usage of relevant statistical tests. We want to elaborate a bit more on the raised criticism of weak evidence: It is true, primarily in the specifically mentioned case of comparing inversion data from models and observations, we cannot rely on strong inter-model relationships to constrain the mediating process of atmospheric stability. The motivation of constraining these highly-defining process to some extent with recent sate-of-the-art climate models is however a tempting option. Connecting to our first reply of this review, we rather show the full story rather than selecting some that match one line of evidence. However, in reflection of all results presented in the conclusions, we focus on what is, from our view, important and conclusive. This concerns two steps of the method: First, identifying model difference where they are clear and consistent across the results, second, attributing co-located observations to either one of the emerging categories (our proposed constraint).

We acknowledge the reviewers comment and elaborate more on data credibility. For the mentioned case that concerns the discussed season of ONDJFM: We argue that the model discrimination (albeit limited by data availability) is supported by there being no overlap in mean inversion strength across the models: During ON for CMIP6/w/s: 4.6–5.8 K / 1.8–3.5 K, and during DJFM for CMIP6/w/s 7.6–10.6 K / 5.8–6.9 K, respectively. A paragraph is added in Section 3.1 L459. This attribution to a specific model subset, not only in the subset average, but also for individual models, is true also for Section 3.2 and 3.3, that use the same models. We further performed a two-sample Kolmogorov-Smirnov test to compare the similarity of CMIP6/w and CMIP6/s distributions (addressed in the same paragraph in Section 3.1). Thereby, we conclude that the first point of model categorisation is fulfilled. Further, primarily during MOSAiC winter, the inversion distribution is most attributable to the range of CMIP6/w models, which is why in the conclusion we highlight the outcome, supported by the two following sections. The shared outcome of Section 3.1–3.3 is now more critically discussed in the discussion.

We further more critically review each individual Section regarding model subset discrimination, and constraints by observation, and further focus more specifically on the synergy of all Sections in the conclusion.

**References**

Christensen, M., Poulsen, C., McGarragh, G., and Grainger, R. (2016). Algorithm Theoretical Basis Document (ATBD) of the Community Code for CLimate (CC4CL) Broadband Radiative Flux Retrieval (CC4CL-TOAFLUX) module - Cloud_CCI Working Group. Technical report, European Space Agency. Last access July 2019.

Clerbaux, N., Russell, J., Dewitte, S., Bertrand, C., Caprion, D., De Paepe, B., Gonzalez Sotelino, L., Ipe, A., Bantges, R., and Brindley, H. (2009). Comparison of GERB instantaneous radiance and flux products with CERES Edition-2 data. *Remote Sensing of Environment*, 113(1):102–114.

Hannachi, A., Straus, D. M., Franzke, C. L., Corti, S., and Woollings, T. (2017). Low-frequency nonlinearity and regime behavior in the northern hemisphere extratropical atmosphere. *Reviews of Geophysics*, 55(1):199–234.

Henderson, D. S., L'Ecuyer, T., Stephens, G., Partain, P., and Sekiguchi, M. (2013). A multisensor perspective on the radiative impacts of clouds and aerosols. *Journal of Applied Meteorology and Climatology*, 52(4):853 – 871.

Kato, S., Rose, F. G., Rutan, D. A., Thorsen, T. J., Loeb, N. G., Doelling, D. R., Huang, X., Smith, W. L., Su, W., and Ham, S.-H. (2018). Surface irradiances of edition 4.0 clouds and the earth's radiant energy system (ceres) energy balanced and filled (ebaf) data product. *Journal of Climate*, 31(11):4501 – 4527.

Kay, J. E. and L'Ecuyer, T. (2013). Observational constraints on Arctic Ocean clouds and radiative fluxes during the early 21st century. *Journal of Geophysical Research: Atmospheres*, 118(13):7219–7236.

Lelli, L., Vountas, M., Khosravi, N., and Burrows, J. P. (2023). Satellite remote sensing of regional and seasonal arctic cooling showing a multi-decadal trend towards brighter and more liquid clouds. *Atmospheric Chemistry and Physics*, 23(4):2579–2611.

Linke, O. and Quaas, J. (2022). The impact of $co_2$-driven climate change on the arctic atmospheric energy budget in cmip6 climate model simulations. *Tellus A: Dynamic Meteorology and Oceanography*, 74(2022).

Loeb, N. G., Doelling, D. R., Wang, H., Su, W., Nguyen, C., Corbett, J. G., Liang, L., Mitrescu, C., Rose, F. G., and Kato, S. (2018). Clouds and the earth's radiant energy system (ceres) energy balanced and filled (ebaf) top-of-atmosphere (toa) edition-4.0 data product. *Journal of Climate*, 31(2):895 – 918.

McGarragh, G. R., Poulsen, C. A., Thomas, G. E., Povey, A. C., Sus, O., Stapelberg, S., Schlundt, C., Proud, S., Christensen, M. W., Stengel, M., Hollmann, R., and Grainger, R. G. (2018). The community cloud retrieval for climate (cc4cl) – part 2: The optimal estimation approach. *Atmospheric Measurement Techniques*, 11(6):3397–3431.

Nakamura, N. and Oort, A. H. (1988). Atmospheric heat budgets of the polar regions. *Journal of Geophysical Research: Atmospheres*, 93(D8):9510–9524.

Palmer, T. N. (1993). Extended-range atmospheric prediction and the lorenz model. *Bulletin of the American Meteorological Society*, 74(1):49–66.

Palmer, T. N. (1999). A nonlinear dynamical perspective on climate prediction. *Journal of Climate*, 12(2):575–591.

Philipp, D., Stengel, M., and Ahrens, B. (2020). Analyzing the Arctic Feedback Mechanism between Sea Ice and Low-Level Clouds Using 34 Years of Satellite Observation. *Journal of Climate*, 33(17):7479 – 7501.

Rantanen, M., Karpechko, A. Y., Lipponen, A., Nordling, K., Hyvärinen, O., Ruosteenoja, K., Vihma, T., and Laaksonen, A. (2022). The arctic has warmed nearly four times faster than the globe since 1979. *Communications Earth & Environment*, 3(1):168.

Serreze, M. C., Barrett, A. P., Slater, A. G., Steele, M., Zhang, J., and Trenberth, K. E. (2007). The large-scale energy budget of the arctic. *Journal of Geophysical Research: Atmospheres*, 112(D11).

Stengel, M., Stapelberg, S., Sus, O., Finkensieper, S., Würzler, B., Philipp, D., Hollmann, R., Poulsen, C., Christensen, M., and McGarragh, G. (2020). Cloud_cci Advanced Very High Resolution Radiometer post meridiem (AVHRR-PM) dataset version 3: 35-year climatology of global cloud and radiation properties. *Earth System Science Data*, 12(1):41–60.

Stengel, M., Stapelberg, S., Sus, O., Schlundt, C., Poulsen, C., Thomas, G., Christensen, M., Carbajal Henken, C., Preusker, R., Fischer, J., Devasthale, A., Willén, U., Karlsson, K.-G., McGarragh, G. R., Proud, S., Povey, A. C., Grainger, R. G., Meirink, J. F., Feofilov, A., Bennartz, R., Bojanowski, J. S., and Hollmann, R. (2017). Cloud property datasets retrieved from AVHRR, MODIS, AATSR and MERIS in the framework of the Cloud_cci project. *Earth System Science Data*, 9(2):881–904.

Sus, O., Stengel, M., Stapelberg, S., McGarragh, G., Poulsen, C., Povey, A. C., Schlundt, C., Thomas, G., Christensen, M., Proud, S., Jerg, M., Grainger, R., and Hollmann, R. (2018). The community cloud retrieval for climate (cc4cl) – part 1: A framework applied to multiple satellite imaging sensors. *Atmospheric Measurement Techniques*, 11(6):3373–3396.

Trenberth, K. E. (1997). Using Atmospheric Budgets as a Constraint on Surface Fluxes. *J. Climate*, 10(11):2796 – 2809.

Vinjamuri, K. S., Vountas, M., Lelli, L., Stengel, M., Shupe, M. D., Ebell, K., and Burrows, J. P. (2023). Validation of the cloud_cci cloud products in the arctic [preprint]. *Atmospheric Measurement Techniques Discussions*, pages 1–24.

Zygmuntowska, M., Mauritsen, T., Quaas, J., and Kaleschke, L. (2012). Arctic Clouds and Surface Radiation - a critical comparison of satellite retrievals and the ERA-Interim reanalysis. *Atmospheric Chemistry and Physics*, 12(14):6667–6677.

---

## Author Response (AR2)

**Editor's comment**

Thank you for submitting a revised version of the manuscript to ACP. I have received two evaluation reports from the original referees. While both referees agree that most of the previous comments are addressed and the manuscript is clearly improved, there are remaining concerns. A major one shared by them both is the potential uncertainty in conclusions imposed by internal variability. The observations and CMIP6 model results are from different time periods, during which the modes of internal variability can be different. Also, how well can the CMIP6/w and CMIP6/s groups of models reproduce the observed internal variability in the analyzed historical period? How will the claimed model biases change if the effect of internal variability is removed from the CMIP6 models? The referees also raised a couple of other major issues. Please refer to their reports. These will need to be addressed before I can make a recommendation for the publication of your manuscript in ACP.

**Reply**: We thank the editor and reviewers for their constructive comments which have helped us to improve the manuscript. In the revised manuscript we now put a new emphasis on the role of internal variability. It has been criticised before that by using the mean of realizations from each participating CMIP6 model alone might not justify 1) interpreting the differences between CMIP6/w and CMIP6/s subsets (weak and strong AA/ALRF models in the historical period), and 2) constraining the climate relevant parameters by observations. We want to give a general remark on these points, before addressing the specific comments of the reviewers below:

   1) By taking the average of model realizations over the past decades, we average out the effect of internal variability, and isolate the response to external forcing. As such, the differences between CMIP6/w and CMIP6/s can be attributed to external forcing. The observations, however, represent a single climate trajectory and thus combine both the effect of internal variability and response to external forcing. We revised the method section that elaborates on the CMIP6 simulations accordingly (newly added Section 2.9 in the manuscript). When comparing the observations to the model subsets (CMIP6/w and CMIP6/s) it is thus important to discuss if the attribution to either one (in our work, based on their respective distributions) can be justified if accounting for internal variability. This leads to the second point:

2) We now discuss our results concerning thermodynamic structure of the boundary layer (i.e., inversion), energy transport, and TOA energy budget (OLR) also in context of internal variability. Specifically, we examine whether the differences between observations, and CMIP6/w / CMIP6/s models, can be explained by internal variability within each subset. In particular, we compute the difference in parameters (OBS minus CMIP/w / CMIP6/s) and compare that difference to the respective range of model realizations which is attributable to internal variability. This range is calculated by subtracting the ensemble mean from each realization (to remove the forced response), and then calculating the central 95 % range of internal variability per model subset. If the OBS-model difference lies within (without) that range, it can (cannot) with confidence be explained by internal variability, which justifies verifying (falsifying) the specific subset based on the OBS. We specify these points within the reviewer's comments and revised the manuscript accordingly.

**Author's response to RC1**

**Major**

**Reviewer Point P 0.1** — The authors stick with the use of short time series as climatological averages, disregarding internal variability. Especially for the MOSAiC winter, that has been shown to have a particular large-scale circulation with less meridional advection of warm air then other recent winter, I do not think this is an appropriate choice.

**Reply**: We thank the reviewer for bringing up internal variability and agree that it should be accounted for. Regarding MOSAiC, the observations during winter are roughly consistent with the ensemble mean of CMIP6/w. This leads to the conclusion that the ensemble mean of CMIP6/w models (response to external forcing) more realistically represents the OBS. We acknowledge, however, the reviewer's concern that the observations might be a low-probability trajectory of the climate system, and therefore need to be put into context of the envelope of model realizations.

We show in Fig. R 1 (also added to the manuscript as Fig. B1) the averages of inversion during DJFM, observed and simulated (ensemble averages), corresponding to Fig. 4 of the manuscript. We also indicate the residuals after subtracting the CMIP6/w and CMIP6/s data from the observations. The error bars account for internal variability of the respective model subset. They are computed by subtracting the subset ensemble mean from each realization, and then calculating the central 95 % range (e.g. England et al., 2021).

[Figure]

Figure R 1: Averaged inversion during DJFM for MOSAiC, CMIP6/w and CMIP/s (ensemble means, respectively), corresponding to Fig. 4 of the manuscript. Gray bars give the residuals after subtracting the externally forced simulation from the observed inversion. The error bars indicate the 95 % range of simulated internal variability of both CMIP/w (blue), and CMIP6/s (red) models, respectively.

The difference between observations and CMIP6/w is small compared to the one for CMIP6/s (this result is already discussed in the manuscript and has led us to the conclusion that CMIP6/w models more realistically represent the inversion). However, individual CMIP6/s realizations might still be consistent with the observed inversion. Fig. R 1 shows that this is not the case: the MOSAiC − CMIP6/s difference cannot be explained with confidence by internal variability of the CMIP6/s ensemble, as is the case for CMIP6/w. This justifies our main conclusion that CMIP6/s models systematically underestimate the inversion. We added these results to the manuscript in L507ff (a similar analysis is also done for atmospheric energy transport and OLR at TOA in Fig. B1 of the manuscript).

The second point of the reviewer concerns the usage of MOSAiC data in general, as it might represent anomalous inversion conditions. It is true that during MOSAiC the Polarstern experienced certain anomalous events, e.g., extreme cases of warm, moist air transported from the northern North Atlantic or northwestern Siberia during late fall until early spring. Rinke et al. (2021) compared the near-surface meteorological conditions during MOSAiC to the context of the recent climatology (characterised by co-located ERA5 reanalyses with hourly resolution 1979–2020). They show that for the full time series, the near-surface meteorological variables were mostly within the record, even during storms and moisture intrusion events. We want to emphasise in particular that this is true for the near-surface air temperature. In order to respond in depth to the reviewer comment, we examined whether this statement also is true for the inversion strength. The result is shown in Fig. R 2, namely a comparison between MOSAiC inversion time series and co-located ERA5 data as statistics of the 30 years preceding MOSAiC (1991–2020). Note that the MOSAiC inversion appears generally smaller than in the manuscript, since we had to interpolate the radiosonde data to common pressure levels of ERA5 (CMIP6 models have pressure data, ERA5 not).

[Figure]

Figure R 2: Inversion strengths $\Delta T$ obtained from radio soundings and concurrent 2-m temperature measurements from the nearby ice camp during MOSAiC, and for ERA5. ERA5 inversion data is computed as the difference between the maximum temperature below the 250 hPa isobar, and the surface (as for CMIP6 and MOSAiC). MOSAiC data is interpolated to ERA5 pressure levels. The blue line shows MOSAiC, and the orange line ERA5 data (inter-annual average 1991–2020). Grey lines give the 5th and 95th percentiles, black lines the minimum–maximum range from 1991–2020 data from ERA5, respectively. Blue and green crosses give the DJFM average value for MOSAiC and ERA5, respectively.

In conclusion, the inversion strength observed during MOSAIC was not unusual. Extending the results of Rinke et al. (2021), it is evident that the MOSAIC inversion lies within the climatological range. In particular, the seasonal average during DJFM is close to the average values from the past years, which justifies the comparison between climate models and MOSAiC data. Another line of evidence is that the average winter-time inversion during MOSAiC is fundamentally similar to the average winter inversion during the SHEBA campaign (approx. 8 K in the averaged DJF temperature profile; Stramler et al., 2011). In addition (albeit not relevant for DJFM), recent work of Svensson et al. (2023) shows that for the MOSAiC April (where most of the warm air intrusion events were recorded), the 2-m temperature from observations and ERA5 are in large agreement for most of the month. We added a comment to the manuscript (L484ff).

**Reviewer Point P 0.2** — It is unclear to me what the profiles over sea ice ($> 15\%$) and their trends actually show. My understanding of the definition is that they would include grid points passing from 100 to 30 % sea-ice cover between the reference and recent period, and thus an important amount of sea-ice retreat, which the authors attempt to exclude from this analysis.

**Reply**: The reviewer is exactly right: where the sea ice concentration is 15 % or higher, the area is considered ice-covered; where sea ice concentration is below 15 %, the area is considered ice-free.

With this definition we follow the recommendation of the NSIDC (https://nsidc.org/data/soac/sea-ice-concentration), and it allows to classify sea ice conditions and their changes. We define sea ice as areas with SIC of >15 % in both reference and warmer climate, open ocean with SIC of <15 % in both reference and warmer climate, and sea-ice retreat as SIC of >15 % in reference climate and<15 % in warmer climate, respectively (e.g. Lauer et al., 2020; Boeke et al., 2021; Linke and Quaas, 2022). What we are actually interested in (Fig. 2 of the manuscript) is the effect of both, surface type (i.e. difference sea ice vs. ocean profile) and cloudiness (overcast and non-overcast).

By comparing, e.g., the two black lines with squared markers in each panel of Fig. 2, we account for different cloud conditions over an equal surface type (sea ice). This allows to isolate the cloud effect at least partly, and its changes in panel c. By comparing, e.g., the solid black and red line (square and triangle markers, respectively), we compare profiles over sea ice and ocean, respectively, at equal cloud conditions (overcast). This aims to isolate the effect of the surface type on the temperature profile, and its changes in panel c. We do not account explicitly for the profile over sea ice retreat, but it is implied by the surface-type difference, and the way it changes.

In order to address the reviewer's comment, we now adapted the caption of Fig. 2 in the revised manuscript in order to clearly explain the above statements, and further expanded the text (L80ff).

**Reviewer Point P 0.3** — The authors conclude that "local processes mediating the lower thermodynamic structure of the atmosphere are more realistically depicted in climate models with weak simulated ALRF/AA in the past". However, in my view the authors have not actually studied the representation of these processes in any subset of models, let alone to the extent to generalize such conclusions. The differences in inversion strengths shown in the manuscript may well be due to different combinations of compensating or non-compensating biases in the underlying processes (mixed-phase clouds, turbulence, sea-ice concentration and thickness, heat conduction through snow and ice).

**Reply**: The reviewer has a good point that this formulation was misleading. What we intended to do is summarise the results of Section 2.2–2.4 (inversion and profiling), and 2.5–2.6. (energy transport). The former conclusions cover the vertical thermodynamic structure of the lower troposphere, whereas the latter can also impact the free troposphere. We agree that "processes mediating" these features is only partly correct, since there are other processes that can impact them, as rightly pointed out. In response to the reviewer's comment, we now adapt the formulation to clarify that we constrain the climate-relevant parameters that relate to the processes, without excluding other impacts: It is now e.g., "Local, near-surface features like temperature inversion", or "the vertical temperature structure of the Arctic boundary layer". We also added subsection 3.2 and 3.3 to more clearly sort local vs. remote features that are linked to AA/ALRF.

**Author's response to RC1**

**Minor**

**Reviewer Point P 0.4** — The authors addressed most of my comments, although they didn't reduce the comparisons to fewer but more robust metrics. For that reason, I am hesitant to recommend "accept". For example, a major, remaining concern is the use of OLR data. As recognised by

[Figure]

Figure R 3: Comparison between OLR data from CERES and AVHRR. Left: Distribution of OLR at TOA. Right: Overlapping time series during 2001–2014. All values are derived as Pan-Arctic and seasonal averages during DJFM.

the authors in their reply, there are significant uncertainties in the observation datasets, obscuring the determination of a trend signal from it. This uncertainty, together with the uncertainty from matching the time periods (against internal variability in the models), makes it very questionable to use the data to discriminate the linear trends in the GCMs. I'd suggest again the authors think more critically about their use of the different data.

**Reply**: We thank the reviewer for their comments regarding credibility of observational data, and the role of internal variability. To address the first point regarding credibility of the OBS we added a second satellite from NOAA/NCEI HIRS, just as ERA5 reanalyses data to the previously used AVHRR record in response to the reviewer's concern in the first review. All three datasets support our conclusions. The combined observational estimate is now derived as the average of these three data records (BEST COMB; added to the article). We further added uncertainty ranges to the trends, which are computed as standard deviation of trends following Lelli et al. (2023). In addition, we compared the AVHRR record with the current standard and that is CERES EBAF 4.2 first edition, published on January 27, 2023 (`https://ceres.larc.nasa.gov/documents/DQ_summaries/CERES_EBAF_Ed4.2_DQS.pdf`). CERES data has not been used in the manuscript due to insufficient time coverage, but it is widely used for data evaluation. For the available overlap years, for latitudes north of 66° N and during boreal winter (DJFM), Fig. R 3 shows the OLR distribution (left), and the time series of the two records (right), respectively. Although some small differences can be detected in the distributions (mainly due to surface characterisation, e.g. emissivity), the consistency of the two time series is further confirmation of the robustness of the records, and the soundness of the derived trend data.

The second point addresses the valid concern regarding the role of internal variability. So far, all model-to-OBS/reanalysis comparisons rely on ensemble averages in the climate model data, i.e., internal variability has been averaged out. The observations, however, comprise only one possible climate trajectory reflecting both the response to external forcing as well as internal climate variability, and therefore need to be put into context of the envelope of model realizations. We revised the manuscript and now discuss our main results (temperature inversion, energy transport, and OLR at TOA) also in the context of internal climate variability (see new method Section 2.9 in the manuscript). To address the specific comment of the reviewer, we show in Fig. R 4 the averages of the OLR anomaly trend during DJFM, observed and simulated (ensemble averages), corresponding to Fig. 9 of the manuscript. We also indicate the residuals after subtracting the CMIP6/w and CMIP6/s data from the BEST COMB (best combined estimate from satellite observations and ERA5). The error bars account for internal variability of the respective model subset. They are computed by subtracting the subset ensemble mean

from each realization, and then calculating the central 95 % ranges (England et al., 2021).

[Figure]

(a) OLR trend anomaly w.r.t. 1983–1994 (DJFM)   (b) OLR average 2000–2014 (DJFM)

Figure R 4: a) Averaged OLR anomaly trend during DJFM for BEST COMB (average from NOAA/NCEI HIRS, ERA5, and AVHRR), CMIP6/w and CMIP/s (ensemble means, respectively), corresponding to Fig. 9 of the manuscript. Gray bars give the residuals after subtracting the externally forced simulation (ensemble means) from the BEST COMB. The error bars indicate the 95 % ranges which could be explained by internal variability per subset. b) Same as a), but for climatological OLR averages 2000–2014.

Fig. R 4a indicates that the difference between BEST COMB and CMIP6/s is smaller compared to CMIP6/w (this result is already discussed in the manuscript, and it has led us to the conclusion that the mean of CMIP6/s simulations more realistically represents the observed OLR trends, albeit still underestimating them). However, also the CMIP6/w ensemble members might still be consistent with the observed OLR trends. Fig. R 4 now allows to conclude that this is unlikely. The fact that the BEST-COMB − CMIP6/s difference is within the range of internal variability simulated by CMIP6/s, but the BEST-COMB − CMIP6/w difference cannot be fully explained by the range of internal variability simulated by CMIP6/w, justifies our previous conclusion. We further show the absolute values of OLR at the end of the historical period (2000–2014), with a similar result: The difference BEST-COMB − CMIP6/s is small compared to OBS − CMIP6/w, and the differences are covered by the range of simulated internal variability for CMIP6/s, but not for CMIP6/w. Fig. R 4a is now added to the manuscript (as Fig. B1), together with a similar analysis of MOSAiC inversion and Pan-Arctic energy transport, to support the main conclusions of our study with regards to the role of internal variability.

**References**

Boeke, R. C., Taylor, P. C., and Sejas, S. A. (2021). On the nature of the arctic's positive lapse-rate feedback. *Geophysical Research Letters*, 48(1):e2020GL091109.

England, M. R., Eisenman, I., Lutsko, N. J., and Wagner, T. J. (2021). The recent emergence of arctic amplification. *Geophysical Research Letters*, 48(15):e2021GL094086.

Lauer, M., Block, K., Salzmann, M., and Quaas, J. (2020). Co2-forced changes of arctic temperature lapse-rates in cmip5 models. *Met. Z.*, 29(1):79–93.

Lelli, L., Vountas, M., Khosravi, N., and Burrows, J. P. (2023). Satellite remote sensing of regional and seasonal arctic cooling showing a multi-decadal trend towards brighter and more liquid clouds. *Atmospheric Chemistry and Physics*, 23(4):2579–2611.

Linke, O. and Quaas, J. (2022). The impact of $co_2$-driven climate change on the arctic atmospheric energy budget in cmip6 climate model simulations. *Tellus A: Dynamic Meteorology and Oceanography*, 74(2022).

Rinke, A., Cassano, J. J., Cassano, E. N., Jaiser, R., and Handorf, D. (2021). Meteorological conditions during the mosaic expedition: Normal or anomalous? *Elem Sci Anth*, 9(1):00023.

Stramler, K., Genio, A. D. D., and Rossow, W. B. (2011). Synoptically driven arctic winter states. *Journal of Climate*, 24(6):1747 – 1762.

Svensson, G., Murto, S., Shupe, M., Pithan, F., Magnusson, L., Day, J., Doyle, J., Renfrew, I., Spengler, T., and Vihma, T. (2023). Warm air intrusions reaching the mosaic expedition in april 2020—the yopp targeted observing period (top). *Elem Sci Anth*, 11.

---

## Author Response (AR3)

**Author's response to RC1**

We thank Referee 1 for their suggestions. Our responses are listed below.

**Major**

**Reviewer Point P 0.1** — The treatment of internal variability legitimately relies on the assumption that models adequately represent internal variability, I would recommend stating that explicitly.

**Reply**: We have added the recommended statement to Section 2.9 which elaborates on the treatment of internal variability (L418).

**Reviewer Point P 0.2** — More importantly, is the internal variability computed by considering each ensemble member and each winter within that ensemble member separately, or is the entire period 2010–2014 considered as one realisation? Only considering each year separately would do justice to the fact that the observational time series only covers one winter. Either way, I recommend explaining this more explicitly in the manuscript.

**Reply**: We acknowledge the reviewer's comment that the treatment of the model period (including several years in the historical simulations) is not explained sufficiently. It is as the reviewer says, each year in the comparing model period is considered separately when deriving the range across model realisations as an estimate of simulated internal variability. We added the explanation to the figure caption in Appendix B.

**Reviewer Point P 0.3** — l. 416f: It might be more precise to say that differences between the subsets are due to the inter-model differences in the response to forcing (as the forcing should be the same across all models)

**Reply**: Agreed, we adapted the sentence accordingly.